# Bringing robotics taxonomies to continuous domains via GPLVM on hyperbolic manifolds

## Abstract

Robotic taxonomies have appeared as high-level hierarchical abstractions that classify how humans move and interact with their environment. They have proven useful to analyse grasps, manipulation skills, and whole-body support poses. Despite the efforts devoted to design their hierarchy and underlying categories, their use in application fields remains scarce. This may be attributed to the lack of computational models that fill the gap between the discrete hierarchical structure of the taxonomy and the high-dimensional heterogeneous data associated to its categories. To overcome this problem, we propose to model taxonomy data via hyperbolic embeddings that capture the associated hierarchical structure. To do so, we formulate a Gaussian process hyperbolic latent variable model and enforce the taxonomy structure through graph-based priors on the latent space and distance-preserving back constraints. We test our model on the whole-body support pose taxonomy to learn hyperbolic embeddings that comply with the original graph structure. We show that our model properly encodes unseen poses from existing or new taxonomy categories, it can be used to generate trajectories between the embeddings, and it outperforms its Euclidean counterparts.

## 1 Introduction

Roboticists are often inspired by biological insights to create robotic systems that exhibit human- or animal-like capabilities (Siciliano & Khatib, 2016). In particular, it is first necessary to understand how humans move and interact with their environment to then generate biologically-inspired motions and behaviors of robotics hands, arms or humanoids. In this endeavor, researchers proposed to structure and categorize human hand postures and body poses into hierarchical classifications known as *taxonomies*. Their structure depends on the variables considered to categorize human motions and their interactions with the environment, as well as on associated qualitative measures.

Different taxonomies have been proposed in the area of human and robot grasping (Cutkosky, 1989; Feix et al., 2016; Abbasi et al., 2016; Stival et al., 2019). Feix et al. (2016) introduced a taxonomy of hand grasps whose structure was mainly defined by the hand pose and the type of contact with the object. Later, Stival et al. (2019) claimed that the taxonomy designed in (Feix et al., 2016) heavily depended on subjective qualitative measures, and proposed a quantitative tree-like taxonomy of hand grasps based on muscular and kinematic patterns. A similar data-driven approach was used to design a grasp taxonomy based on sensed contact forces in (Abbasi et al., 2016). Robotic manipulation also gave rise to various taxonomies. Bullock et al. (2013) introduced a hand-centric manipulation taxonomy that classifies manipulation skills according to the type of contact with the objects and the object motion imparted by the hand. A different strategy was developed in (Paulius et al., 2019), where a manipulation taxonomy was designed based on a categorization of contacts and motion trajectories. Humanoid robotics also made significant efforts to analyze human motions, thus proposing taxonomies as high-level abstractions of human motion configurations. Borràs et al. (2017) analyzed the contacts of the human limbs with the environment and designed a taxonomy of whole-body support poses.

In addition to being used for analysis purposes in robotics or biomechanics, some of the aforementioned taxonomies were leveraged for modeling grasp actions (Romero et al., 2010; Lin & Sun, 2015), for planning contact-aware whole-body pose sequences (Mandery et al., 2016), and for learning manipulation skills embeddings (Paulius et al., 2020). However, despite most taxonomies carry

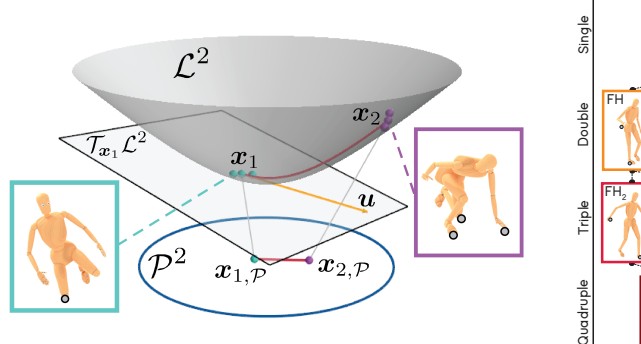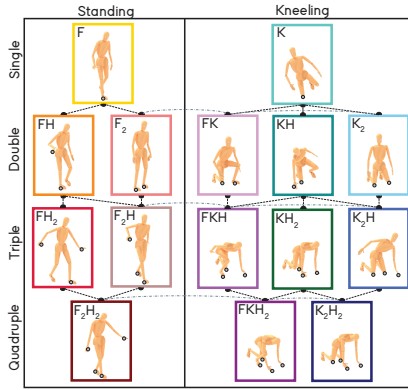

Figure 1: *Left:* Illustration of the Lorentz $\mathcal{L}^2$ and Poincaré $\mathcal{P}^2$ models of the hyperbolic manifold. The former is depicted as the gray hyperboloid, while the latter is represented by the blue circle. Both models show a geodesic (——) between two points $\boldsymbol{x}_1$ (•) and $\boldsymbol{x}_2$ (•). The vector $\boldsymbol{u}$ (——) lies on the tangent space of $\boldsymbol{x}_1$ such that $\boldsymbol{u} = \mathrm{Log}_{\boldsymbol{x}_1}(\boldsymbol{x}_2)$. *Right:* Subset of the whole-body support pose taxonomy (Borràs et al., 2017) used in our experiments. Each node is a support pose defined by the type of contacts (foot F, hand H, knee K). The lines represent graph transitions between the taxonomy nodes. Contacts are depicted by grey dots.

a well-defined hierarchical structure, it is often overlooked. First, these taxonomies are usually exploited for classification tasks whose target classes are mainly the tree leaves, disregarding the full taxonomy structure (Feix et al., 2016; Abbasi et al., 2016). Second, the discrete representation of the taxonomy categories hinders their use for motion generation (Romero et al., 2010).

We believe that the difficulty of leveraging robotic taxonomies is due to the lack of computational models that exploit *(i)* the domain knowledge encoded in the hierarchy, and *(ii)* the information of the high-dimensional data associated to the taxonomy categories. We tackle this problem from a representation learning perspective by modeling taxonomy data as embeddings that capture the associated hierarchical structure. Inspired by recent advances on word embeddings (Nickel & Kiela, 2017; 2018; Mathieu et al., 2019), we propose to leverage the *hyperbolic manifold* (Ratcliffe, 2019) to learn such embeddings. An important property of the hyperbolic manifold is that distances grow exponentially when moving away from the origin, and shortest paths between distant points tend to pass through it, resembling a *continuous hierarchical structure*. Therefore, we hypothesize that the geometry of the hyperbolic manifold allows us to learn embeddings that comply with the original graph structure of robotic taxonomies.

Specifically, we propose a Gaussian process hyperbolic latent variable model (GPHLVM) to learn embeddings of taxonomy data on the hyperbolic manifold. To do so, we impose a hyperbolic geometry to the latent space of the well-known GPLVM (Lawrence, 2003; Titsias & Lawrence, 2010). This demands to reformulate the Gaussian distribution, the kernel, and the optimization process of the vanilla GPLVM to account for the geometry of the hyperbolic latent space. To do so, we leverage the hyperbolic wrapped Gaussian distribution (Nagano et al., 2019), and provide a positive-definite-guaranteed approximation of the hyperbolic kernel proposed by McKean (1970). Moreover, we resort to Riemannian optimization (Absil et al., 2007; Boumal, 2022) to optimize the GPHLVM parameters. We enforce the taxonomy graph structure in the learned embeddings through graph-based priors on the latent space and via graph-distance-preserving back constraints (Lawrence & Quiñonero Candela, 2006; Urtasun et al., 2008). Our GPHLVM is conceptually similar to the GPLVM for Lie groups introduced in (Jensen et al., 2020), which also imposes geometric properties to the GPLVM latent space. However, our formulation is specifically designed for the hyperbolic manifold and fully built on tools from Riemannian geometry. Moreover, unlike (Tosi et al., 2014) and (Jørgensen & Hauberg, 2021), where the latent space was endowed with a pullback Riemannian metric learned via the GPLVM mapping, we impose the hyperbolic geometry to the GPHLVM latent space as an inductive bias adapted to our targeted applications.

We test our approach on graphs extracted from the whole-body support pose taxonomy (Borràs et al., 2017). The proposed GPHLVM learns hyperbolic embeddings of the body support poses that comply with the original graph structure, and properly encodes unseen poses from existing or new taxonomy nodes. Moreover, we show how we can exploit the continuous geometry of the hyperbolic manifold to generate trajectories between different embeddings pairs, which comply with the taxonomy graph structure. To the best of our knowledge, this paper is the first to leverage the hyperbolic manifold for robotic applications.

## 2 BACKGROUND

**Gaussian Process Latent Variable Models:** A GPLVM defines a generative mapping from latent variables $\{\boldsymbol{x}_n\}_{n=1}^N, \boldsymbol{x}_n \in \mathbb{R}^Q$ to observations $\{\boldsymbol{y}_n\}_{n=1}^N, \boldsymbol{y}_n \in \mathbb{R}^D$ by modeling the corresponding non-linear transformation with Gaussian processes (GPs) (Lawrence, 2003). The GPLVM is described as

$$y_{n,d} \sim \mathcal{N}(y_{n,d}; f_{n,d}, \sigma_d^2) \quad \text{with} \quad f_{n,d} \sim \text{GP}(m_d(\boldsymbol{x}_n), k_d(\boldsymbol{x}_n, \boldsymbol{x}_n)) \quad \text{and} \quad \boldsymbol{x}_n \sim \mathcal{N}(\boldsymbol{0}, \boldsymbol{I}), \quad (1)$$

where $y_{n,d}$ denotes the $d$-th dimension of the observation $\boldsymbol{y}_n$, $m_d(\cdot) : \mathbb{R}^Q \mapsto \mathbb{R}$ and $k_d(\cdot, \cdot) : \mathbb{R}^Q \times \mathbb{R}^Q \to \mathbb{R}$ are the GP mean and kernel function, respectively, and $\sigma_d^2$ is a hyperparameter. Classically, the hyperparameters and latent variables of the GPLVM were optimized using *maximum likelihood* or *maximum a posteriori* (MAP) estimates. As this does not scale gracefully to large datasets, contemporary methods use inducing points and variational approximations of the evidence (Titsias & Lawrence, 2010). Compared to neural-network-based generative models, GPLVMs are data efficient and provide automatic uncertainty quantification.

**Riemannian geometry:** To understand the hyperbolic manifold, it is necessary to first define some basic Riemannian geometry concepts (Lee, 2018). To begin with, consider a Riemannian manifold $\mathcal{M}$, which is a locally Euclidean topological space with a globally-defined differential structure. For each point $\boldsymbol{x} \in \mathcal{M}$, there exists a tangent space $\mathcal{T}_{\boldsymbol{x}}\mathcal{M}$ that is a vector space consisting of the tangent vectors of all the possible smooth curves passing through $\boldsymbol{x}$. A Riemannian manifold is equipped with a Riemannian metric, which permits to define curve lengths in $\mathcal{M}$. Shortest-path curves, called geodesics, can be seen as the generalization of straight lines on the Euclidean space to Riemannian manifolds, as they are minimum-length curves between two points in $\mathcal{M}$. To operate with Riemannian manifolds, it is common practice to exploit the Euclidean tangent spaces. To do so, we resort to mappings back and forth between $\mathcal{T}_{\boldsymbol{x}}\mathcal{M}$ and $\mathcal{M}$, which are the exponential and logarithmic maps. The exponential map $\text{Exp}_{\boldsymbol{x}}(\boldsymbol{u}) : \mathcal{T}_{\boldsymbol{x}}\mathcal{M} \to \mathcal{M}$ maps a point $\boldsymbol{u}$ in the tangent space of $\boldsymbol{x}$ to a point $\boldsymbol{y}$ on the manifold, so that it lies on the geodesic starting at $\boldsymbol{x}$ in the direction $\boldsymbol{u}$, and such that the geodesic distance $d_{\mathcal{M}}$ between $\boldsymbol{x}$ and $\boldsymbol{y}$ equals the distance between $\boldsymbol{x}$ and $\boldsymbol{u}$. The inverse operation is the logarithmic map $\text{Log}_{\boldsymbol{x}}(\boldsymbol{u}) : \mathcal{M} \to \mathcal{T}_{\boldsymbol{x}}\mathcal{M}$. Finally, the parallel transport $\text{P}_{\boldsymbol{x} \to \boldsymbol{y}}(\boldsymbol{u}) : \mathcal{T}_{\boldsymbol{x}}\mathcal{M} \to \mathcal{T}_{\boldsymbol{y}}\mathcal{M}$ operates with manifold elements lying on different tangent spaces.

**Hyperbolic manifold:** The hyperbolic space $\mathbb{H}^d$ is the unique simply-connected complete $d$-dimensional Riemannian manifold with a constant negative sectional curvature (Ratcliffe, 2019). There are several isometric models for the hyperbolic space, in particular, the Poincaré ball $\mathcal{P}^d$ and the Lorentz (hyperboloid) model $\mathcal{L}^d$ (see Fig. 1-*left*). The latter representation is chosen here as it is numerically more stable than the former, and thus better suited for Riemannian optimization. However, the Poincaré model provides a more intuitive representation and is here used for visualization. This is easily achieved by leveraging the isometric mapping between both models (see App. A for details). An important property of the hyperbolic manifold is the exponential rate of the volume growth of a ball with respect to its radius. In other words, distances in $\mathbb{H}^d$ grow exponentially when moving away from the origin, and shortest paths between distant points on the manifold tend to pass through the origin, resembling a continuous hierarchical structure. Because of this, the hyperbolic manifold is often exploited to embed hierarchical data such as trees or graphs (Nickel & Kiela, 2017; Chami et al., 2020). Although its potential to embed discrete data structures into a continuous space is well known in the machine learning community, its application in robotics is presently scarce.

**Hyperbolic wrapped Gaussian distribution:** Probabilistic models on Riemannian manifolds demand to have probability distributions that consider the manifold geometry. We use the hyperbolic wrapped distribution (Nagano et al., 2019), which builds on a Gaussian distribution on the tangent space at the origin $\boldsymbol{\mu}_0 = (1, 0, \ldots, 0)^\mathsf{T}$ of $\mathbb{H}^d$, that is then projected onto the hyperbolic space after transporting the tangent space to the desired location. Intuitively, the construction of this wrapped distribution is as follows: *(1)* sample a point $\tilde{\boldsymbol{v}} \in \mathbb{R}^d$ from the Euclidean normal distribution $\mathcal{N}(\boldsymbol{0}, \boldsymbol{\Sigma})$, *(2)* transform $\tilde{\boldsymbol{v}}$ to an element of $\mathcal{T}_{\boldsymbol{\mu}_0}\mathbb{H}^d \subset \mathbb{R}^{d+1}$ by setting $\boldsymbol{v} = (0, \tilde{\boldsymbol{v}})^\mathsf{T}$, *(3)* apply the parallel transport $\boldsymbol{u} = \text{P}_{\boldsymbol{\mu}_0 \to \boldsymbol{\mu}}(\boldsymbol{v})$, and *(4)* project $\boldsymbol{u}$ to $\mathbb{H}^d$ via $\text{Exp}_{\boldsymbol{\mu}}(\boldsymbol{u})$. The resulting probability density function is

$$\log \mathcal{N}_{\mathbb{H}^d}(\boldsymbol{x}; \boldsymbol{\mu}, \boldsymbol{\Sigma}) = \log \mathcal{N}(\boldsymbol{v}; \boldsymbol{0}, \boldsymbol{\Sigma}) - (d-1) \log\left( \sinh(\|\boldsymbol{u}\|_{\mathcal{L}}) / \|\boldsymbol{u}\|_{\mathcal{L}} \right), \quad (2)$$

where $\boldsymbol{v} = \text{P}_{\boldsymbol{\mu} \to \boldsymbol{\mu}_0}(\boldsymbol{u})$, $\boldsymbol{u} = \text{Log}_{\boldsymbol{\mu}}(\boldsymbol{x})$, and $\|\boldsymbol{u}\|_{\mathcal{L}} = \sqrt{\langle \boldsymbol{u}, \boldsymbol{u} \rangle_{\boldsymbol{\mu}}}$. The hyperbolic wrapped distribution (Nagano et al., 2019) has a more general expression given in (Skopek et al., 2020).

## 3 GAUSSIAN PROCESS HYPERBOLIC LATENT VARIABLE MODEL

We present the proposed GPHLVM, which extends GPLVM to hyperbolic latent spaces. A GPHLVM defines a generative mapping from the hyperbolic latent space $\mathbb{H}^Q$ to the observation space, e.g. the data associated to the taxonomy, based on GPs. By considering independent GPs across the observation dimensions, the GPHLVM is formally described as

$$y_{n,d} \sim \mathcal{N}(y_{n,d}; f_{n,d}, \sigma_d^2) \quad \text{with} \quad f_{n,d} \sim \text{GP}(m_d(\boldsymbol{x}_n), k_d^{\mathbb{H}^Q}(\boldsymbol{x}_n, \boldsymbol{x}_n)) \quad \text{and} \quad \boldsymbol{x}_n \sim \mathcal{N}_{\mathbb{H}^Q}(\boldsymbol{\mu}_0, \alpha\boldsymbol{I}), \tag{3}$$

where $y_{n,d}$ denotes the $d$-th dimension of the observation $\boldsymbol{y}_n \in \mathbb{R}^D$ and $\boldsymbol{x}_n \in \mathbb{H}^Q$ is the corresponding latent variable. Our GPHLVM is built on hyperbolic GPs, characterized by a mean function $m_d(\cdot) : \mathbb{H}^Q \to \mathbb{R}$ (usually set to 0), and a kernel $k_d^{\mathbb{H}^Q}(\cdot, \cdot) : \mathbb{H}^Q \times \mathbb{H}^Q \to \mathbb{R}$. These kernels encode similarity information in the latent hyperbolic manifold and should reflect its geometry to perform effectively, as detailed in §. 3.1. Also, the latent variable $\boldsymbol{x} \in \mathbb{H}^Q$ is assigned a hyperbolic wrapped Gaussian prior $\mathcal{N}_{\mathbb{H}^Q}(\boldsymbol{\mu}_0, \alpha\boldsymbol{I})$ based on Eq. 2, where $\boldsymbol{\mu}_0$ is the origin of $\mathbb{H}^Q$, and the parameter $\alpha$ controls the spread of the latent variables in $\mathbb{H}^Q$. As Euclidean GPLVMs, our GPHLVM can be trained by finding a MAP estimate or via variational inference. However, special care must be taken to guarantee that the latent variables belong to the hyperbolic manifold, as explained in §. 3.2.

### 3.1 HYPERBOLIC KERNELS

For GPs in Euclidean spaces, the squared exponential (SE) and Matérn kernels are standard choices (Rasmussen & Williams, 2006). In the modern machine learning literature these were generalized to non-Euclidean spaces such as manifolds (Borovitskiy et al., 2020; Jaquier et al., 2021) or graphs (Borovitskiy et al., 2021). The generalized SE kernels may be connected to the much studied *heat kernels*. These are given (cf. Grigoryan & Noguchi (1998)) by

$$k^{\mathbb{H}^2}(\boldsymbol{x}, \boldsymbol{x}') = \frac{\sigma^2}{C_\infty} \int_\rho^\infty \frac{s e^{-s^2/(2\kappa^2)}}{(\cosh(s) - \cosh(\rho))^{1/2}} \, \mathrm{d}s, \quad k^{\mathbb{H}^3}(\boldsymbol{x}, \boldsymbol{x}') = \frac{\sigma^2}{C_\infty} \frac{\rho}{\sinh \rho} e^{-\rho^2/(2\kappa^2)}, \tag{4}$$

where $\rho = \text{dist}_{\mathbb{H}^d}(\boldsymbol{x}, \boldsymbol{x}')$ denotes the geodesic distance between $\boldsymbol{x}, \boldsymbol{x}' \in \mathbb{H}^d$, $\kappa$ and $\sigma^2$ are the kernel lengthscale and variance, and $C_\infty$ is a normalizing constant. To the best of our knowledge, no closed form expression for $\mathbb{H}^2$ is known. To approximate the kernel in this case, a discretization of the integral is performed. One appealing option is the Monte Carlo approximation based on the truncated Gaussian density. Unfortunately, such approximations easily fail to be positive semidefinite if the number of samples is not very large. We address this via an alternative Monte Carlo approximation

$$k^{\mathbb{H}^2}(\boldsymbol{x}, \boldsymbol{x}') \approx \frac{\sigma^2}{C_\infty'} \frac{1}{L} \sum_{l=1}^{L} s_l \tanh(\pi s_l) e^{(2s_l i + 1)\langle \boldsymbol{x}_\mathcal{P}, \boldsymbol{b}_l \rangle} \overline{e^{(2s_l i + 1)\langle \boldsymbol{x}'_\mathcal{P}, \boldsymbol{b}_l \rangle}}, \tag{5}$$

where $\langle \boldsymbol{x}_\mathcal{P}, \boldsymbol{b} \rangle = \frac{1}{2} \log \frac{1 - |\boldsymbol{x}_\mathcal{P}|^2}{|\boldsymbol{x}_\mathcal{P} - \boldsymbol{b}|^2}$ is the hyperbolic outer product with $\boldsymbol{x}_\mathcal{P}$ being the representation of $\boldsymbol{x}$ as a point on the Poincaré disk $\mathcal{P}^2 = \mathbb{D}$, $i, \overline{z}$ denote the imaginary unit and complex conjugation, respectively, $\boldsymbol{b}_l \overset{\text{i.i.d.}}{\sim} U(\mathbb{T})$ with $\mathbb{T}$ the unit circle, and $s_l \overset{\text{i.i.d.}}{\sim} e^{-s^2\kappa^2/2} \mathbb{1}_{[0,\infty)}(s)$. The distributions of $\boldsymbol{b}_l$ and $s_l$ are easy to sample from: The former is sampled by applying $x \to e^{2\pi i x}$ to $x \sim U([0,1])$ and the latter is (proportional to) a truncated normal distribution. Importantly, the right-hand side of Eq. 5 is easily recognized to be an inner product in the space $\mathbb{C}^L$, which immediately implies its positive semidefiniteness (see App. B for the development of Eq. 5). Note that hyperbolic kernels for $\mathbb{H}^Q$ with $Q > 3$ are generally defined as integrals of the kernels Eq. 4 (Grigoryan & Noguchi, 1998). Analogs of Matérn kernels for $\mathbb{H}^Q$ are obtained as integral of the SE kernel of the same dimension (Jaquier et al., 2021).

### 3.2 MODEL TRAINING

Similarly to the Euclidean case, training the GPHLVM is equivalent to finding an optimal set of latent variables $\{\boldsymbol{x}_n\}_{n=1}^N$ and hyperparameters $\boldsymbol{\Theta} = \{\theta_d\}_{d=1}^D$, with $\boldsymbol{x}_n \in \mathbb{H}^Q$ and $\theta_d$ the hyperparameters of the $d$-th GP. For small datasets, the GPHLVM can be trained by maximizing the log posterior of the model, i.e., $\mathcal{L}_{\text{MAP}} = \log\left(p(\boldsymbol{Y}|\boldsymbol{X})p(\boldsymbol{X})\right)$ with $\boldsymbol{Y} = (\boldsymbol{y}_1 \dots \boldsymbol{y}_N)^\mathsf{T}$ and $\boldsymbol{X} = (\boldsymbol{x}_1 \dots \boldsymbol{x}_N)^\mathsf{T}$. For large datasets, the GPHLVM can be trained, similarly to the so-called Bayesian GPLVM (Titsias & Lawrence, 2010), by maximizing the marginal likelihood of the data,

i.e., $\mathcal{L}_{\text{MaL}} = \log p(\boldsymbol{Y}) = \log \int p(\boldsymbol{Y}|\boldsymbol{X})p(\boldsymbol{X})d\boldsymbol{X}$. As this quantity is intractable, it is approximated via variational inference by adapting the methodology introduced in (Titsias & Lawrence, 2010) to hyperbolic latent spaces, as explained next.

**Variational inference:** We approximate the posterior $p(\boldsymbol{X}|\boldsymbol{Y})$ by a variational distribution $q(\boldsymbol{X})$ defined as a hyperbolic wrapped normal distribution over the latent variables, i.e.,

$$q_\phi(\boldsymbol{X}) = \prod_{n=1}^{N} \mathcal{N}_{\mathbb{H}^Q}(\boldsymbol{x}_n; \boldsymbol{\mu}_n, \boldsymbol{\Sigma}_n), \tag{6}$$

with variational parameters $\phi = \{\boldsymbol{\mu}_n, \boldsymbol{\Sigma}_n\}_{n=1}^N$, with $\boldsymbol{\mu}_n \in \mathbb{H}^Q$ and $\boldsymbol{\Sigma}_n \in \mathcal{T}_{\boldsymbol{\mu}_n}\mathbb{H}^Q$. Similarly to the Euclidean case (Titsias & Lawrence, 2010), this variational distribution allows the formulation of a lower bound

$$\log p(\boldsymbol{Y}) \geq \mathbb{E}_{q_\phi(\boldsymbol{X})}\left[\log p(\boldsymbol{Y}|\boldsymbol{X})\right] - \text{KL}\big(q_\phi(\boldsymbol{X})||p(\boldsymbol{X})\big). \tag{7}$$

The KL divergence $\text{KL}\big(q_\phi(\boldsymbol{X})||p(\boldsymbol{X})\big)$ between two hyperbolic wrapped normal distributions can easily be evaluated via Monte-Carlo sampling (see App. C.1 for details). Moreover, the expectation $\mathbb{E}_{q_\phi(\boldsymbol{X})}\left[\log p(\boldsymbol{Y}|\boldsymbol{X})\right]$ can be decomposed into individual terms for each observation dimension as $\sum_{d=1}^D \mathbb{E}_{q_\phi(\boldsymbol{X})}\left[\log p(\boldsymbol{y}_d|\boldsymbol{X})\right]$, where $\boldsymbol{y}_d$ is the $d$-th column of $\boldsymbol{Y}$. For large datasets, each term can be evaluated via a variational sparse GP approximation (Titsias, 2009; Hensman et al., 2015). To do so, we introduce $M$ inducing inputs $\{\boldsymbol{z}_{d,m}\}_{m=1}^M$ with $\boldsymbol{z}_{d,m} \in \mathbb{H}^Q$ for each observation dimension $d$, whose corresponding inducing variables $\{u_{d,m}\}_{m=1}^M$ are defined as noiseless observations of the GP in Eq. 3, i.e, $u_d \sim \text{GP}(m_d(\boldsymbol{z}_d), k_d^{\mathbb{H}^Q}(\boldsymbol{z}_d, \boldsymbol{z}_d))$. Similar to (Hensman et al., 2015), we can write

$$\log p(\boldsymbol{y}_d|\boldsymbol{X}) \geq \mathbb{E}_{q_\lambda(\boldsymbol{f}_d)}\left[\log \mathcal{N}(\boldsymbol{y}_d; \boldsymbol{f}_d(\boldsymbol{X}), \sigma_d^2)\right] - \text{KL}\big(q_\lambda(\boldsymbol{u}_d)||p(\boldsymbol{u}_d|\boldsymbol{Z}_d)\big), \tag{8}$$

where we defined $q_\lambda(\boldsymbol{f}_d) = \int p(\boldsymbol{f}_d|\boldsymbol{u}_d)q_\lambda(\boldsymbol{u}_d)d\boldsymbol{u}_d$ with the variational distribution $q_\lambda(\boldsymbol{u}_d) = \mathcal{N}(\boldsymbol{u}_d; \tilde{\boldsymbol{\mu}}_d, \tilde{\boldsymbol{\Sigma}}_d)$ and variational parameters $\lambda = \{\tilde{\boldsymbol{\mu}}_d, \tilde{\boldsymbol{\Sigma}}_d\}_{d=1}^D$. Remember that the inducing variables $u_{d,m}$ are Euclidean, i.e., the variational distribution $q_\lambda(\boldsymbol{u}_d)$ is a Euclidean Gaussian and the KL divergence in Eq. 8 has a closed-form solution. In this case, the training parameters of the GPHLVM are the set of inducing inputs $\{\boldsymbol{z}_{d,m}\}_{m=1}^M$, the variational parameters $\phi$ and $\lambda$, and the hyperparameters $\boldsymbol{\Theta}$ (see App. C.2 for the full derivation of the GPHLVM variational inference process).

**Optimization:** As several training parameters of the GPHLVM belong to $\mathbb{H}^Q$, i.e., the latent variables $\boldsymbol{x}_n$ for the MAP estimation, or the inducing inputs $\boldsymbol{z}_{d,m}$ and means $\boldsymbol{\mu}_n$ for variational inference. To account for the hyperbolic geometry of these parameters, we leverage Riemannian optimization methods (Absil et al., 2007; Boumal, 2022) to train the GPHLVM. Each step of first order (stochastic) Riemannian optimization methods is generally of the form

$$\boldsymbol{\eta}_t \leftarrow h\big(\text{grad}\,\mathcal{L}(\boldsymbol{x}_t), \boldsymbol{\tau}_{t-1}\big), \qquad \boldsymbol{x}_{t+1} \leftarrow \text{Exp}_{\boldsymbol{x}_t}(-\alpha_t\boldsymbol{\eta}_t), \qquad \boldsymbol{\tau}_t \leftarrow \text{P}_{\boldsymbol{x}_t \to \boldsymbol{x}_{t+1}}\big(\boldsymbol{\eta}_t\big). \tag{9}$$

The update $\boldsymbol{\eta}_t \in \mathcal{T}_{\boldsymbol{x}_t}\mathcal{M}$ is first computed as a function $h$ of the Riemannian gradient grad of the loss $\mathcal{L}(\boldsymbol{x}_t)$ and of $\boldsymbol{\tau}_{t-1}$, the previous update parallel-transported to the tangent space of the new estimate $\boldsymbol{x}_t$. The estimate $\boldsymbol{x}_t$ is then updated by projecting the update $\boldsymbol{\eta}_t$ scaled by a learning rate $\alpha_t$ onto the manifold using the exponential map. The function $h$ is equivalent to computing the update of the Euclidean algorithm, e.g., $\boldsymbol{\eta}_t \leftarrow \text{grad}\,\mathcal{L}(\boldsymbol{x}_t)$ for a simple gradient descent. Notice that Eq. 9 is applied on a product of manifolds when optimizing several parameters. In this paper, we used the Riemannian Adam (Bécigneul & Ganea, 2019) implemented in Geoopt (Kochurov et al., 2020) to optimize the GPHLVM parameters.

## 4 INCORPORATING TAXONOMY KNOWLEDGE INTO GPHLVM

While we are now able to learn hyperbolic embeddings of the data associated to a taxonomy using our GPHLVM, they do not necessarily follow the graph structure of the taxonomy. In other words, the manifold distances between pairs of embeddings do not need to match the graph distances. To overcome this, we introduce graph-distance information as inductive bias to learn the embeddings. To do so, we leverage two well-known techniques in the GPLVM literature: priors on the embeddings and back constraints (Lawrence & Quiñonero Candela, 2006; Urtasun et al., 2008). Both are reformulated to preserve the taxonomy graph structure in the hyperbolic latent space as a function of the node-to-node shortest paths.

**Graph-distance priors:** As shown by Urtasun et al. (2008), the structure of the latent space can be modified by adding priors of the form $p(\boldsymbol{X}) \propto e^{-\phi(\boldsymbol{X})/\sigma_\phi^2}$ to the GPLVM, where $\phi(\boldsymbol{X})$ is a function that we aim at minimizing. Incorporating such a prior may also be understood as augmenting the GPLVM loss $\mathcal{L}$ with a regularization term $-\phi(\boldsymbol{X})$. Therefore, we propose to augment the loss of the GPHLVM with a distance-preserving graph-based regularizer. Several such losses have been proposed in the literature, see (Cruceru et al., 2021) for a review. Specifically, we define $\phi(\boldsymbol{X})$ as the stress loss

$$\mathcal{L}_{\text{stress}}(\boldsymbol{X}) = \sum_{i<j} \big( \text{dist}_{\mathbb{G}}(c_i, c_j) - \text{dist}_{\mathbb{H}^Q}(\boldsymbol{x}_i, \boldsymbol{x}_j) \big)^2, \tag{10}$$

where $c_i$ denotes the taxonomy node to which the observation $\boldsymbol{y}_i$ belongs, and $\text{dist}_{\mathbb{G}}, \text{dist}_{\mathbb{H}^Q}$ are the taxonomy graph distance and the geodesic distance on $\mathbb{H}^Q$, respectively. The loss Eq. 10 encourages the preservation of all distances of the taxonomy graph in $\mathbb{H}^Q$. It therefore acts *globally*, thus allowing the complete taxonomy structure to be reflected by the GPHLVM. Notice that Cruceru et al. (2021) also survey a distortion loss that encourages the distance of the embeddings to match the graph distance by considering their ratio. We notice, however, that this distortion loss is only properly defined when the embeddings $\boldsymbol{x}_i$ and $\boldsymbol{x}_j$ correspond to different classes $c_i \neq c_j$. Interestingly, our empirical results using this loss were lackluster and numerically unstable (see App. E).

**Back-constraints:** The back-constrained GPLVM (Lawrence & Quiñonero Candela, 2006) defines the latent variables as a function of the observations, i.e., $x_{n,q} = g_q(\boldsymbol{y}_1 \ldots, \boldsymbol{y}_n, \boldsymbol{w}_q)$ with parameters $\{\boldsymbol{w}_q\}_{q=1}^Q$. It allows us to incorporate new observations in the latent space after training, while preserving local similarities between observations in the embeddings. To incorporate graph-distance information into the GPHLVM and ensure that latent variables lie on the hyperbolic manifold, we propose the back-constraints mapping

$$\boldsymbol{x}_n = \text{Exp}_{\boldsymbol{\mu}_0}(\tilde{\boldsymbol{x}}_n) \quad \text{with} \quad \tilde{x}_{n,q} = \sum_{m=1}^N w_{q,m} k^{\mathbb{R}^D}(\boldsymbol{y}_n, \boldsymbol{y}_m) k^{\mathbb{G}}(c_n, c_m). \tag{11}$$

The mapping Eq. 11 not only expresses the similarities between data in the observation space via the kernel $k^{\mathbb{R}^J}$, but encodes the relationships between data belonging to nearby taxonomy nodes via $k^{\mathbb{G}}$. In other words, similar observations associated to the same (or near) taxonomy nodes will be close to each other in the resulting latent space. The kernel $k^{\mathbb{G}}$ is a Matérn kernel on the taxonomy graph following the formulation introduced in (Borovitskiy et al., 2021), which accounts for the graph geometry (see also App. D). We also use a Euclidean SE kernel for $k^{\mathbb{R}^D}$. Notice that the back constraints only incorporate *local* information into the latent embedding. Therefore, to preserve the *global* graph structure, we pair the proposed back-constrained GPHLVM with the stress prior Eq. 10. Note that both kernels are required in Eq. 11: By defining the mapping as a function of the graph kernel only, the observations of each taxonomy node would be encoded by a single latent point. When using the observation kernel only, dissimilar observations of the same taxonomy node would be distant in the latent space, despite the additional stress prior, as $k^{\mathbb{R}^D}(\boldsymbol{y}_n, \boldsymbol{y}_m) \approx 0$.

## 5 EXPERIMENTS

We test the proposed GPHLVM to model data of the whole-body support pose taxonomy (Borràs et al., 2017). Each node of the taxonomy graph (see Fig. 1-*right*) is a support pose defined by its contacts, so that the distance between nodes can be viewed as the number of contact changes required to go from a support pose to another. We use standing and kneeling poses of the datasets in (Mandery et al., 2016) and (Langenstein, 2020). The former were extracted from recordings of a human walking without hand support, or using supports from a handrail or from a table on one side or on both sides. The latter were obtained from a human standing up from a kneeling position. Each pose is identified with a node of the graph of Fig. 1-*right*. We test our approach on three different datasets: an *unbalanced* dataset (i.e., 100 poses composed of 72 standing and 28 kneeling poses); a *balanced* dataset (i.e., only 60 standing poses); and an *joint-space* dataset (i.e., same 60 standing poses represented as joint configurations). For the first two datasets each pose is represented as a vector $\boldsymbol{y}_n = [\boldsymbol{y}_{\text{LF}}, \boldsymbol{y}_{\text{RF}}, \boldsymbol{y}_{\text{LH}}, \boldsymbol{y}_{\text{RH}}]^\top \in \mathbb{R}^{12}$ corresponding to the positions of the human's feet and hands. Instead, for the last dataset, each pose is represented by vector of joint angles $\boldsymbol{y}_n \in \mathbb{R}^{44}$. Last but not least, we also test our approach on an augmented version of the whole-body support pose taxonomy, which explicitly distinguishes between left and right contacts. The main results are analyzed in the sequel, while additional experimental details and results are given in App. F and G.

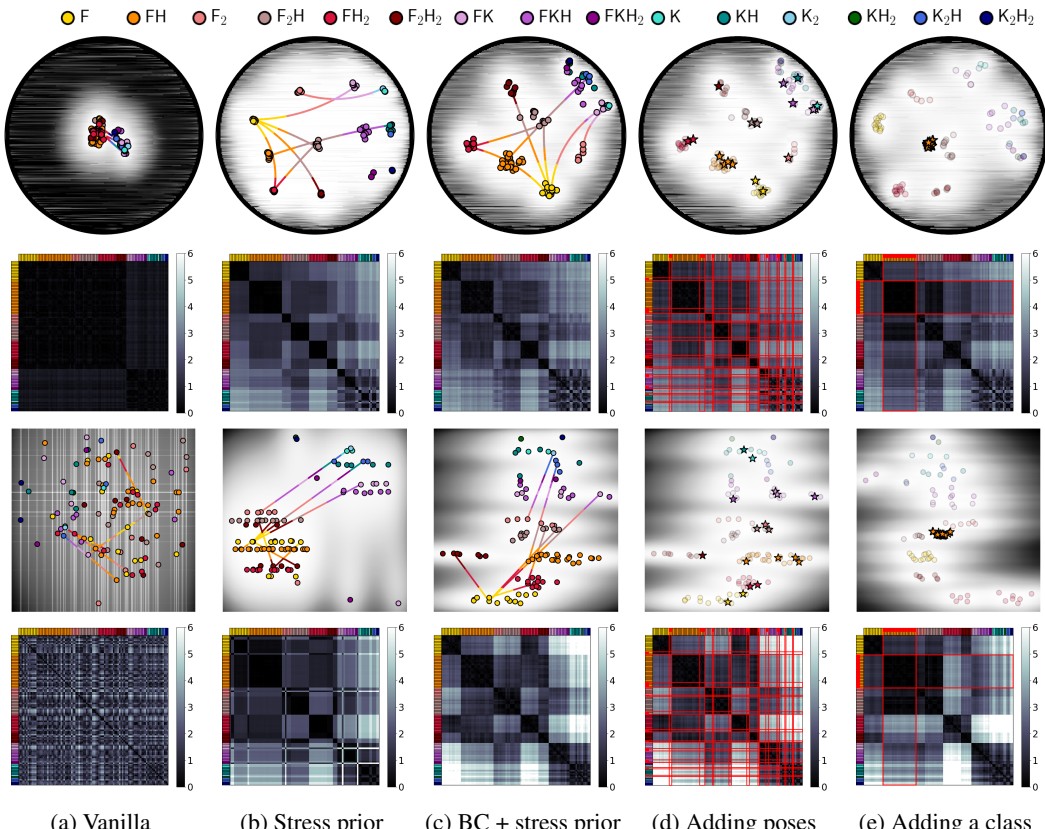

Figure 2: The first and last two rows respectively show the latent embeddings and examples of interpolating geodesics in $\mathcal{P}^2$ and $\mathbb{R}^2$, followed by pairwise distance matrices. Embeddings colors match those of Fig. 1-*right*, and background colors indicate the GPLVM uncertainty. Added poses *(d)* and classes *(e)* are marked with stars and highlighted with red in the distance matrices.

**Hyperbolic embeddings of support poses:** We embed the 100 standing and kneeling poses into 2-dimensional hyperbolic and Euclidean spaces using GPHLVM and GPLVM. For each, we test the model without regularization, with stress prior, and with back-constraints coupled with stress prior (see App. F.2 for the training parameters). Figs. 2a-2c show the learned embeddings alongside distance matrices, which are to be compared with the graph distances in Fig. 3. As shown in Fig. 2a, the models without regularization do not encode any meaningful distance structure in latent space. In contrast, the models with stress prior result in embeddings that comply with the taxonomy graph structure: The embeddings are grouped and organized according to the taxonomy nodes, the geodesic distances match the graph ones, and arguably more so in the hyperbolic case (see Figs. 2b-2c). This is further reflected in the stress values of the latent embeddings with respect to the graph distances (see Table 1). Interestingly, the hyperbolic models also outperform Euclidean models with 3-dimensional latent spaces (see App. G.1).

| | Regularization | Stress $\pm\sigma$ |
|---|---|---|
| $\mathbb{R}^2$ | No reg. | 2.15±2.92 |
| | Stress | 0.86±2.18 |
| | BC+Stress | 1.70±2.91 |
| | — " —: unseen poses | 0.53 ± 0.86 |
| | — " —: unseen class | 0.85±1.0 |
| $\mathbb{H}^2$ | No reg. | 3.71±4.08 |
| | Stress | **0.14**±0.20 |
| | BC+Stress | **0.21**±0.34 |
| | — " —: unseen poses | **0.22**±0.34 |
| | — " —: unseen class | **0.58**±0.65 |

Table 1: Average stress per geometry and regularization.

This is due to the fact that the geometry of the hyperbolic manifold leads to exponentially-increasing distances w.r.t the origin, which provides an increased volume to match the graph structure when compared to Euclidean spaces, thus resulting in better low-dimensional representations of taxonomy data. Our GPHLVM also outperformed vanilla and hyperbolic versions of variational autoencoders (VAE) to encode meaningful taxonomy information in the latent space (see App. G.4). In general, the tested VAEs only captured a global structure that separates standing from kneeling poses. Moreover, the average stress of the VAEs' latent embeddings is higher compared to the GPHLVM's. Finally, notice that the back constraints further or-

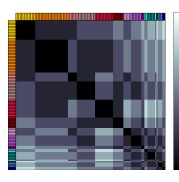

Figure 3: Graph distance between the poses following Fig. 1-*right*.

ganize the embeddings inside a class according to the similarity between their observations (Fig. 2c).

**Taxonomy expansion and unseen poses encoding:** An advantage of back-constrained GPLVMs is their affordance to "embed" new observations into the latent space. We test the GPHLVM ability

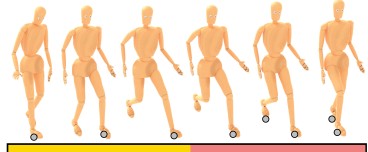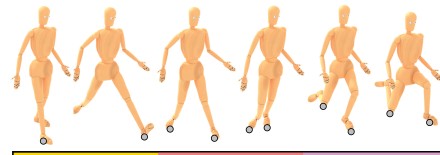

Figure 4: Motions obtained via geodesic interpolation in the back-constrained GPHLVM latent space. *Left*: F to $F_2$. *Right*: F to FK. The colorbars identify the support pose of the closest pose in the latent space.

to place unseen poses or taxonomy classes into the latent space, hypothesizing that their respective embeddings would be placed at meaningful distances w.r.t. the rest of the latent points. First, we consider a back-constrained GPHLVM with stress prior previously trained on example poses from the taxonomy (i.e., the model of Fig. 2c) and embedded unseen poses. Fig. 2d shows how these new poses land close to their respective class cluster. Second, we train a new GPHLVM while withholding all poses corresponding to the $F_1H_1$ class. We then encode these poses and find that they are located at sensible distance when compared to the model trained on the full dataset. Although this is accomplished by both models, the GPHLVM displays lower stress values (see Table 1).

**Trajectory generation via geodesics:** The geometry of the GPHLVM latent space can also be exploited to generate trajectories in the latent space by following the geodesic, i.e. the shortest path, between two embeddings. In other words, our GPHLVM intrinsically provides a mechanism to plan motions via geodesics in the low-dimensional latent space. Examples of geodesics between two poses are shown in Figs. 2b-2c, with the colors along the trajectory matching the class corresponding to the closest hyperbolic latent point. Importantly, the geodesics in our GPHLVM latent space follow the transitions between classes defined in the taxonomy. In other words, the shortest paths in the hyperbolic embedding correspond to the shortest paths in the taxonomy graph. For instance, the geodesic from F to $F_2H_2$ follows $F \rightarrow F_2 \rightarrow F_2H \rightarrow F_2H_2$, while the geodesic from FH to $K_2H$ follows $FH \rightarrow F_2H \rightarrow FKH \rightarrow KH \rightarrow K_2H$. In contrast, straight lines in the Euclidean embeddings often do not match the graph shortest path, resulting in transitions that do not exist in the taxonomy, e.g., $F \rightarrow F_2H_2$, or $F_2 \rightarrow FKH$ in the Euclidean latent space of Figs. 2b-2c (see also App. F.4).

Fig. 4 shows examples of motions resulting from geodesic interpolation in the GPHLVM latent space. As expected, the resulting trajectories do not correspond to direct interpolations between the given initial and final poses. This is due to the lack of information about the objects location and the type of contact in the considered poses. Therefore, poses with very different feet and hands positions may belong to the same class, e.g., two-feet contact with a left hand contact on the handrail or a right hand contact on the table both belong to $F_2H$. This results in artifacts throughout the interpolations, which are alleviated by augmenting the taxonomy to differentiate between left and right contacts, as described next. However, it is interesting that the motions are still consistent with the observed transitions, e.g., the hand positions vary little along a path involving only foot and knee contacts.

**Augmented taxonomy for enhanced trajectory generation:** Here, we aim at improving the quality of the generated motion by augmenting the whole-body support pose taxonomy with additional contact information. To do so, we consider an *augmented* whole-body support pose taxonomy which explicitly distinguishes between left and right contacts by adapting the nodes and transitions of Fig. 1-*right*. For instance, the 1-foot contact (F) node is separated into left-foot ($F^l$) and right-foot ($F^r$) contact nodes. To facilitate motion planning and to test the GPHLVM ability of dealing with high-dimensional spaces, we represent each pose as a vector $\boldsymbol{y}_n \in \mathbb{R}^{44}$ of joint angles instead of a vector of hands and feet positions. A video of the resulting motions accompanies this paper.

We embed the 60 standing poses described in App. G.2 into 3-dimensional hyperbolic and Euclidean spaces using GPHLVM and GPLVM, respectively. For each approach, we test the model without regularization, with stress prior, and with back-constraints coupled with stress prior (see App. G.3 and F.2 for detailed results and training parameters). Fig. 5 shows examples of motions planned by following geodesics in the GPHLVM latent space. We observe that the motions generated by considering the augmented taxonomy result in more realistic interpolations between the given initial and final poses than the trajectories of Fig. 4. Moreover, the previously-observed artifacts are drastically reduced. This is due to the fact that the augmented taxonomy differentiates between left and right contacts, thus allowing very different poses to be placed far apart in the latent space. For example, poses corresponding to $F^lH^r$ and $F^rH^l$ in the augmented taxonomy belonged to the same FH node in the original taxonomy, and were embedded close together. It is also interesting to notice that considering joint angles instead of end-effector positions results in more realistic poses. Such poses may also be obtained by considering both end-effector positions and orientations as observations, which would require an extension of the GPHLVM to handle observations on Riemannian manifolds.

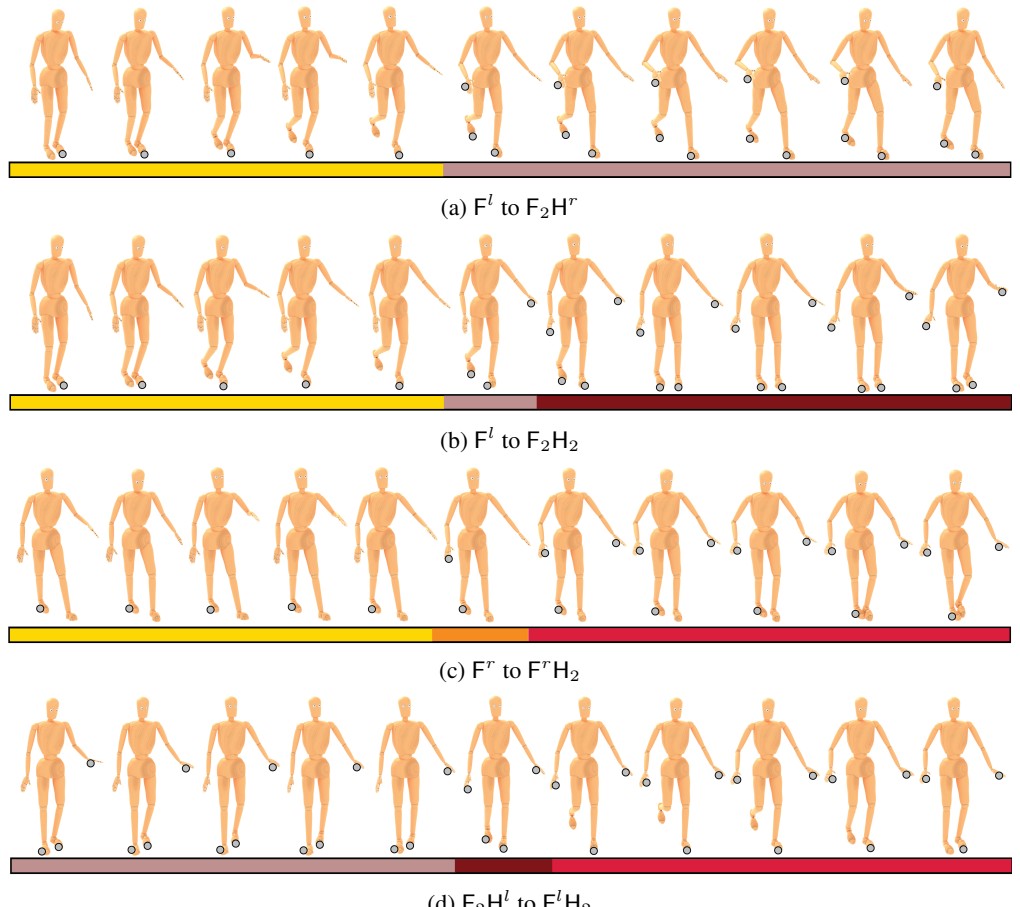

(a) $\mathsf{F}^l$ to $\mathsf{F}_2\mathsf{H}^r$

(b) $\mathsf{F}^l$ to $\mathsf{F}_2\mathsf{H}_2$

(c) $\mathsf{F}^r$ to $\mathsf{F}^r\mathsf{H}_2$

(d) $\mathsf{F}_2\mathsf{H}^l$ to $\mathsf{F}^l\mathsf{H}_2$

Figure 5: Motions obtained via geodesic interpolation in the latent space of the back-constrained GPHLVM trained on the augmented taxonomy (Fig. 10c). Contacts are denoted by gray circles. The colorbars identify the support pose of the closest pose in the latent space.

## 6 CONCLUSIONS

Inspired by the recent developments of taxonomies in different robotics fields, we proposed a computational model GPHLVM that leveraged two types of domain knowledge: the structure of a human-designed taxonomy and a hyperbolic geometry on the latent space which complies with the intrinsic taxonomy's hierarchical structure. Our GPHLVM allows us to learn hyperbolic embeddings of the features of the taxonomy nodes while capturing the associated hierarchical structure. To achieve this, our model exploited the curvature of the hyperbolic manifold and the graph-distance information, as inductive bias. We showed that these two forms of inductive bias are essential to: learn taxonomy-aware embeddings, encode unseen data, and potentially expand the learned taxonomy. Moreover, we reported that vanilla Euclidean approaches underperformed on all the foregoing cases. Finally, we introduced a mechanism to generate taxonomy-aware motions in the hyperbolic latent space.

It is important to emphasize that our geodesic motion generation does not use explicit knowledge on how physically feasible the generated trajectories are. We plan to investigate how to include physics constraints or explicit contact data into the GPHLVM to obtain physically-feasible motions that can be executed on real robots. Moreover, we will work on alleviating the computational cost of the hyperbolic kernel in $\mathbb{H}^d$. This could be tackled by using a different sampling strategy: Instead of sampling from a Gaussian distribution for the approximation Eq. 5, we could sample from the Rayleigh distribution. This is because complex numbers, whose real and imaginary components are i.i.d. Gaussian, have absolute value that is Rayleigh-distributed. As our current experimental study focused on testing our model on different graphs extracted from the whole-body support pose taxonomy (Borràs et al., 2017), we plan to test it with datasets used to design other robotic taxonomies. Finally, we plan to investigate other types of manifold geometries that may accommodate more complex structures coming from highly-heterogeneous graphs (Giovanni et al., 2022).

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

# A  Hyperbolic manifold

## A.1  Equivalence of Poincaré and Lorentz models

As pointed out in the main text (§ 2), it is possible to map points from the Lorentz model to the Poincaré ball via an isometric mapping. Formally, such an isometry is defined as the mapping function $f : \mathcal{L}^d \to \mathcal{P}^d$ such that

$$f(\boldsymbol{x}) = \frac{(x_1, \dots, x_d)^\top}{x_0 + 1}, \tag{12}$$

where $\boldsymbol{x} \in \mathcal{L}^d$ with components $x_0, x_1, \dots, x_d$. The inverse mapping $f^{-1} : \mathcal{P}^d \to \mathcal{L}^d$ is defined as follows

$$f^{-1}(\boldsymbol{y}) = \frac{\left(1 + \|\boldsymbol{y}\|^2, 2y_1, \dots, 2y_d\right)^\top}{1 - \|\boldsymbol{y}^2\|}, \tag{13}$$

with $\boldsymbol{y} \in \mathcal{P}^d$ with components $y_1, \dots, y_d$. Notice that we used the mapping Eq. 12 to represent the hyperbolic embeddings in the Poincaré disk throughout the paper, as well as in the computation of the kernel $k^{\mathbb{H}^2}$ Eq. 4.

## A.2  Manifold operations

As mentioned in the main text (§ 2), we resort to the exponential and logarithmic maps to operate with Riemannian manifold data. The exponential map $\mathrm{Exp}_{\boldsymbol{x}}(\boldsymbol{u}) : \mathcal{T}_{\boldsymbol{x}}\mathcal{M} \to \mathcal{M}$ maps a point $\boldsymbol{u}$ in the tangent space of $\boldsymbol{x}$ to a point $\boldsymbol{y}$ on the manifold, while the logarithmic map $\mathrm{Log}_{\boldsymbol{x}}(\boldsymbol{u}) : \mathcal{M} \to \mathcal{T}_{\boldsymbol{x}}\mathcal{M}$ performs the corresponding inverse operation. In some settings, it is necessary to work with data lying on different tangent spaces of the manifold. In this case, one needs to operate with all data on a single tangent space, which can be achieved by leveraging the parallel transport $\mathrm{P}_{\boldsymbol{x}\to\boldsymbol{y}}(\boldsymbol{u}) : \mathcal{T}_{\boldsymbol{x}}\mathcal{M} \to \mathcal{T}_{\boldsymbol{y}}\mathcal{M}$. All the aforementioned operators are defined in Table 2 for the Lorentz model $\mathcal{L}^d$. Moreover, we introduce the inner product $\langle \boldsymbol{u}, \boldsymbol{v} \rangle_{\boldsymbol{x}}$ between two points on $\mathcal{L}^d$, which is used to compute the geodesic distance $d_{\mathcal{M}}(\boldsymbol{u}, \boldsymbol{v})$ and all the foregoing operations in the Lorentz model, as shown in Table 2.

| Operation | Formula |
|-----------|---------|
| $\langle \boldsymbol{u}, \boldsymbol{v} \rangle_{\boldsymbol{x}}$ | $-u_0 v_0 + \sum_{i=1}^d u_i v_i$ |
| $d_{\mathcal{M}}(\boldsymbol{u}, \boldsymbol{v})$ | $\mathrm{arcosh}(-\langle \boldsymbol{u}, \boldsymbol{v} \rangle_{\boldsymbol{x}})$ |
| $\mathrm{Exp}_{\boldsymbol{x}}(\boldsymbol{u})$ | $\cosh(\|\boldsymbol{u}\|_{\mathcal{L}})\boldsymbol{x} + \sinh(\|\boldsymbol{u}\|_{\mathcal{L}})\frac{\boldsymbol{u}}{\|\boldsymbol{u}\|_{\mathcal{L}}}$ with $\|\boldsymbol{u}\|_{\mathcal{L}} = \sqrt{\langle \boldsymbol{u}, \boldsymbol{u} \rangle_{\boldsymbol{x}}}$ |
| $\mathrm{Log}_{\boldsymbol{x}}(\boldsymbol{y})$ | $\frac{d_{\mathcal{M}}(\boldsymbol{x}, \boldsymbol{y})}{\sqrt{\alpha^2 - 1}}(\boldsymbol{y} + \alpha\boldsymbol{x})$ with $\alpha = \langle \boldsymbol{x}, \boldsymbol{y} \rangle_{\boldsymbol{x}}$ |
| $\mathrm{P}_{\boldsymbol{x}\to\boldsymbol{y}}(\boldsymbol{v})$ | $\boldsymbol{v} + \frac{\langle \boldsymbol{y}, \boldsymbol{v} \rangle_{\boldsymbol{x}}}{1 - \langle \boldsymbol{x}, \boldsymbol{y} \rangle_{\boldsymbol{x}}}(\boldsymbol{x} + \boldsymbol{y})$ |

Table 2: Principal operations on $\mathbb{H}^d$ for the Lorentz model. For more details, see (Bose et al., 2020) and (Peng et al., 2021).

# B  Hyperbolic kernels

As mentioned in the main text (§ 3.1), following the developments on kernels on manifolds like Borovitskiy et al. (2020); Jaquier et al. (2021), we may identify the generalized squared exponential kernel with the *heat kernel*—an important object studied on its own in the mathematical literature. Due to this, we can obtain the expressions Eq. 4. The expression for the case of $\mathbb{H}^2$ requires discretizing the integral, which may lead to an approximation that is not positive semidefinite. We address this by suggesting another approximation guaranteed to be positive semidefinite.

Reversing the derivation in (Chavel, 1984, p. 246), we obtain

$$k^{\mathbb{H}^2}_{\infty, \kappa, \sigma^2}(\boldsymbol{x}, \boldsymbol{x}') = \frac{\sigma^2}{C'_\infty} \int_0^\infty \exp(-s^2/(2\kappa^2)) P_{-1/2+is}(\cosh(\rho)) s \tanh(\pi s) \mathrm{d}s, \tag{14}$$

where $\rho = \mathrm{dist}_{\mathbb{H}^d}(\boldsymbol{x}, \boldsymbol{x}')$ denotes the geodesic distance between $\boldsymbol{x}, \boldsymbol{x}' \in \mathbb{H}^2$, $\kappa$ and $\sigma^2$ are the kernel lengthscale and variance, $C'_\infty$ is a normalizing constant and $P_\alpha$ are Legendre functions Abramowitz & Stegun (1964). Now we prove that these Legendre functions are connected to the *spherical functions* — special functions closely tied to the geometry of the hyperbolic space and possessing a very important property.

**Proposition.** *Assume the disk model of $\mathbb{H}^2$ (i.e. the Poincaré disk). Denote the disk by $\mathbb{D}$ and its boundary, the circle, by $\mathbb{T}$. Define the hyperbolic outer product by $\langle \boldsymbol{z}, \boldsymbol{b} \rangle = \frac{1}{2} \log \frac{1-|\boldsymbol{z}|^2}{|\boldsymbol{z}-\boldsymbol{b}|^2}$ for $\boldsymbol{z} \in \mathbb{D}, \boldsymbol{b} \in \mathbb{T}$. Then*

$$P_{-1/2+is}(\cosh(\rho)) = \underbrace{\int_\mathbb{T} e^{(2si+1)\langle \boldsymbol{z}, \boldsymbol{b} \rangle} \mathrm{d}b}_{\text{spherical function } \phi_{2s}(\boldsymbol{z})} = \int_\mathbb{T} e^{(2si+1)\langle \boldsymbol{z}_1, \boldsymbol{b} \rangle} \overline{e^{(2si+1)\langle \boldsymbol{z}_2, \boldsymbol{b} \rangle}} \mathrm{d}\boldsymbol{b}, \qquad (15)$$

*where $\boldsymbol{z} \in \mathbb{D}$ is such that $\rho = \mathrm{dist}_{\mathbb{H}^2}(\boldsymbol{z}, \boldsymbol{0})$ and $\boldsymbol{z}_1, \boldsymbol{z}_2 \in \mathbb{D}$ are such that $\rho = \mathrm{dist}_{\mathbb{H}^2}(\boldsymbol{z}_1, \boldsymbol{z}_2)$. Here $i$ denotes the imaginary unit and $\overline{\boldsymbol{z}}$ is the complex conjugation.*

*Proof.* Let $\theta$ denote the angle between $\boldsymbol{z}$ and $\boldsymbol{b}$, and note the following simple identities

$$|\boldsymbol{z} - \boldsymbol{b}|^2 = |\boldsymbol{z}|^2 + 1 - 2|\boldsymbol{z}| \cos(\theta) = \tanh(\rho)^2 + 1 - 2\tanh(\rho)\cos(\theta), \qquad (16)$$

$$1 - |\boldsymbol{z}|^2 = 1 - \tanh(\rho)^2 = \cosh(\rho)^{-2}. \qquad (17)$$

Then, we write

$$e^{(2si+1)\langle \boldsymbol{z}, \boldsymbol{b} \rangle} = \left( \frac{|\boldsymbol{z} - \boldsymbol{b}|^2}{1 - |\boldsymbol{z}|^2} \right)^{-si-1/2} = \left( \cosh(\rho)^2 (\tanh(\rho)^2 + 1 - 2\tanh(\rho)\cos(\theta)) \right)^{-si-1/2}, \qquad (18)$$

$$= \left( \sinh(\rho)^2 + \cosh(\rho)^2 - 2\sinh(\rho)\cosh(\rho)\cos(\theta) \right)^{-si-1/2}, \qquad (19)$$

$$= \left( \cosh(2\rho) + \sinh(2\rho)\cos(\theta) \right)^{-si-1/2}. \qquad (20)$$

On the other hand, by (Lebedev et al., 1965, Eq. 7.4.3), we have $P_a(\cosh(x)) = \frac{1}{\pi} \int_0^\pi (\cosh(x) + \sinh(x)\cos(\theta))^a \mathrm{d}\theta$, hence

$$P_{-1/2+is}(\cosh(2\rho)) = \frac{1}{\pi} \int_0^\pi (\cosh(2\rho) + \sinh(2\rho)\cos(\theta))^{-1/2+is} \mathrm{d}\theta, \qquad (21)$$

$$= \frac{1}{2\pi} \int_{-\pi}^\pi (\cosh(2\rho) + \sinh(2\rho)\cos(\theta))^{-1/2+is} \mathrm{d}\theta, \qquad (22)$$

$$= \int_\mathbb{T} e^{(-2si+1)\langle \boldsymbol{z}, \boldsymbol{b} \rangle} \mathrm{d}\boldsymbol{b} = \phi_{-2s}(\boldsymbol{z}). \qquad (23)$$

This computation roughly follows Cohen & Lifshits (2012, Section 4.3.4). Now, by Cohen & Lifshits (2012, Section 3.5), we have $\phi_{-2s}(\boldsymbol{z}) = \phi_{2s}(\boldsymbol{z})$ which proves the first identity. Finally, Lemma 3.5 from Cohen & Lifshits (2012) proves the second identity. $\qquad \square$

By combining expressions Eq. 14 and Eq. 15, we get the following Monte Carlo approximation

$$k_{\infty,\kappa,\sigma^2}^{\mathbb{H}^2}(\boldsymbol{x}, \boldsymbol{x}') \approx \frac{\sigma^2}{C'_\infty} \frac{1}{L} \sum_{l=1}^L s_l \tanh(\pi s_l) e^{(2s_l i+1)\langle \boldsymbol{x}_\mathcal{P}, \boldsymbol{b}_l \rangle} \overline{e^{(2s_l i+1)\langle \boldsymbol{x}'_\mathcal{P}, \boldsymbol{b}_l \rangle}}, \qquad (24)$$

where $\boldsymbol{b}_l \overset{\text{i.i.d.}}{\sim} U(\mathbb{T})$ and $s_l \overset{\text{i.i.d.}}{\sim} e^{-s^2\kappa^2/2} \mathbb{1}_{[0,\infty)}(s)$. This gives the approximation used in the main text (see § 3.1).

Having established a way to evaluate or approximate the heat kernel, analogs of Matérn kernels can be defined by

$$k_{\nu,\kappa,\sigma^2}(\boldsymbol{x}, \boldsymbol{x}') = \frac{\sigma^2}{C_\nu} \int_0^\infty u^{\nu-1} e^{-\frac{2\nu}{\kappa^2} u} \tilde{k}_{\infty,\sqrt{2u},\sigma^2}(\boldsymbol{x}, \boldsymbol{x}') \mathrm{d}u, \qquad (25)$$

where $\tilde{k}_{\infty,\sqrt{2u},\sigma^2}$ is the same as $k_{\infty,\sqrt{2u},\sigma^2}$ but with the normalizing constant $\sigma^2/C_\infty$ dropped for simplicity. Here $C_\nu$ is the normalizing constant ensuring that $k_{\nu,\kappa,\sigma^2}(\boldsymbol{x}, \boldsymbol{x}) = \sigma^2$ for all $\boldsymbol{x}$.

## C  GPHLVM VARIATIONAL INFERENCE

As mentioned in § 3.2, when training our GPHLVM on large datasets, we resort to variational inference as originally proposed in (Titsias & Lawrence, 2010). Here we provide the mathematical details about the changes that are needed to train our model via variational inference.

### C.1  COMPUTING THE KL DIVERGENCE BETWEEN TWO HYPERBOLIC WRAPPED NORMAL DISTRIBUTIONS

As mentioned in § 3.2, we approximate the KL divergence between two hyperbolic wrapped distributions via Monte-Carlo sampling. Namely, given two hyperbolic wrapped distributions $q_\phi(\boldsymbol{x})$ and $p(\boldsymbol{x})$, we write

$$\text{KL}\big(q_\phi(\boldsymbol{x})||p(\boldsymbol{x})\big) = \int q_\phi(\boldsymbol{x}) \log \frac{q_\phi(\boldsymbol{x})}{p(\boldsymbol{x})} d\boldsymbol{x} \approx \frac{1}{K} \sum_{k=1}^{K} \log \frac{q_\phi(\boldsymbol{x}_k)}{p(\boldsymbol{x}_k)}, \tag{26}$$

where we used $K$ independent Monte-Carlo samples drawn from $q_\phi(\boldsymbol{x})$ to approximate the KL divergence. These samples are obtained via the procedure described in § 2, i.e., by sampling an element on the tangent space of the origin $\boldsymbol{\mu}_0 = (1, 0, \ldots, 0)^\mathsf{T}$ of $\mathbb{H}^d$, via a Euclidean normal distribution, and then applying the parallel transport operation and the exponential map to project it onto $\mathbb{H}^d$.

### C.2  DETAILS OF THE VARIATIONAL PROCESS

As mentioned in the main text, the marginal likelihood $p(\boldsymbol{Y})$ is approximated via variational inference by approximating the posterior $p(\boldsymbol{X}|\boldsymbol{Y})$ with the hyperbolic variational distribution $q_\phi(\boldsymbol{X})$ as defined by Eq. 6. The lower bound Eq. 7 is then obtained, similarly as in (Titsias & Lawrence, 2010), as

$$\log p(\boldsymbol{Y}) = \log \int p(\boldsymbol{Y}|\boldsymbol{X})p(\boldsymbol{X})d\boldsymbol{X} \tag{27}$$

$$= \log \int p(\boldsymbol{Y}|\boldsymbol{X})p(\boldsymbol{X})\frac{q_\phi(\boldsymbol{X})}{q_\phi(\boldsymbol{X})}d\boldsymbol{X} = \log \mathbb{E}_{q_\phi(\boldsymbol{X})}\left[\frac{p(\boldsymbol{Y}|\boldsymbol{X})p(\boldsymbol{X})}{q_\phi(\boldsymbol{X})}\right] \tag{28}$$

$$\geq \mathbb{E}_{q_\phi(\boldsymbol{X})}\left[\log \frac{p(\boldsymbol{Y}|\boldsymbol{X})p(\boldsymbol{X})}{q_\phi(\boldsymbol{X})}\right] = \int q_\phi(\boldsymbol{X}) \log \frac{p(\boldsymbol{Y}|\boldsymbol{X})p(\boldsymbol{X})}{q_\phi(\boldsymbol{X})}d\boldsymbol{X} \tag{29}$$

$$= \int q_\phi(\boldsymbol{X}) \log p(\boldsymbol{Y}|\boldsymbol{X})d\boldsymbol{X} - \int q_\phi(\boldsymbol{X}) \log \frac{q_\phi(\boldsymbol{X})}{p(\boldsymbol{X})}d\boldsymbol{X} \tag{30}$$

$$= \mathbb{E}_{q_\phi(\boldsymbol{X})}\left[\log p(\boldsymbol{Y}|\boldsymbol{X})\right] - \text{KL}\big(q_\phi(\boldsymbol{X})||p(\boldsymbol{X})\big), \tag{31}$$

following Jensen's inequality in Eq. 29. As mentioned in § 3.2, the expectation $\mathbb{E}_{q_\phi(\boldsymbol{X})}\left[\log p(\boldsymbol{Y}|\boldsymbol{X})\right]$ can be decomposed into individual terms for each observation dimension as $\sum_{d=1}^{D} \mathbb{E}_{q_\phi(\boldsymbol{X})}\left[\log p(\boldsymbol{y}_d|\boldsymbol{X})\right]$, where $\boldsymbol{y}_d$ is the $d$-th column of $\boldsymbol{Y}$. We then define the inducing inputs $\boldsymbol{Z}_d$ and inducing variables $\boldsymbol{u}_d$ the same way as the noiseless observations $\boldsymbol{f}_d$, so that the joint distribution of $\boldsymbol{f}_d$ and $\boldsymbol{u}_d$ can be written as

$$p(\boldsymbol{f}_d, \boldsymbol{u}_d) = \begin{pmatrix} \boldsymbol{f}_d \\ \boldsymbol{u}_d \end{pmatrix} = \mathcal{N}\left(\begin{pmatrix} \boldsymbol{m}_d(\boldsymbol{X}) \\ \boldsymbol{m}_d(\boldsymbol{Z}_d) \end{pmatrix}, \begin{pmatrix} k_d(\boldsymbol{X}, \boldsymbol{X}) & k_d(\boldsymbol{X}, \boldsymbol{Z}_d) \\ k_d(\boldsymbol{Z}_d, \boldsymbol{X}) & k_d(\boldsymbol{Z}_d, \boldsymbol{Z}_d) \end{pmatrix}\right). \tag{32}$$

The lower bound Eq. 8 is then obtained for each dimension, similarly as in (Hensman et al., 2015), as

$$\log p(\boldsymbol{y}_d|\boldsymbol{X}) = \int \log p(\boldsymbol{y}_d|\boldsymbol{X}, \boldsymbol{u}_d) p(\boldsymbol{u}_d) d\boldsymbol{u}_d \tag{33}$$

$$= \log \int p(\boldsymbol{y}_d|\boldsymbol{X}, \boldsymbol{u}_d) p(\boldsymbol{u}_d) \frac{q_\lambda(\boldsymbol{u}_d)}{q_\lambda(\boldsymbol{u}_d)} d\boldsymbol{u}_d = \log \mathbb{E}_{q_\lambda(\boldsymbol{u}_d)} \left[ \frac{p(\boldsymbol{y}_d|\boldsymbol{X}, \boldsymbol{u}_d) p(\boldsymbol{u}_d)}{q_\lambda(\boldsymbol{u}_d)} \right] \tag{34}$$

$$\geq \mathbb{E}_{q_\lambda(\boldsymbol{u}_d)} \left[ \log \frac{p(\boldsymbol{y}_d|\boldsymbol{X}, \boldsymbol{u}_d) p(\boldsymbol{u}_d)}{q_\lambda(\boldsymbol{u}_d)} \right] = \int q_\lambda(\boldsymbol{u}_d) \log \frac{p(\boldsymbol{y}_d|\boldsymbol{X}, \boldsymbol{u}_d) p(\boldsymbol{u}_d)}{q_\lambda(\boldsymbol{u}_d)} d\boldsymbol{u}_d \tag{35}$$

$$= \int q_\lambda(\boldsymbol{u}_d) \log p(\boldsymbol{y}_d|\boldsymbol{X}, \boldsymbol{u}_d) d\boldsymbol{u}_d - \int q_\lambda(\boldsymbol{u}_d) \log \frac{q_\lambda(\boldsymbol{u}_d)}{p(\boldsymbol{u}_d)} d\boldsymbol{u}_d \tag{36}$$

$$= \mathbb{E}_{q_\lambda(\boldsymbol{u}_d)} \left[ \log p(\boldsymbol{y}_d|\boldsymbol{X}, \boldsymbol{u}_d) \right] - \mathrm{KL}\big(q_\lambda(\boldsymbol{u}_d)||p(\boldsymbol{u}_d)\big) \tag{37}$$

$$\geq \mathbb{E}_{q_\lambda(\boldsymbol{u}_d)} \left[ \mathbb{E}_{p(\boldsymbol{f}_d|\boldsymbol{u}_d)} \left[ \log p(\boldsymbol{y}_d|\boldsymbol{f}_d(\boldsymbol{X})) \right] \right] - \mathrm{KL}\big(q_\lambda(\boldsymbol{u}_d)||p(\boldsymbol{u}_d)\big) \tag{38}$$

$$= \mathbb{E}_{q_\lambda(\boldsymbol{f}_d)} \left[ \log p(\boldsymbol{y}_d|\boldsymbol{f}_d(\boldsymbol{X})) \right] - \mathrm{KL}\big(q_\lambda(\boldsymbol{u}_d)||p(\boldsymbol{u}_d|\boldsymbol{Z}_d)\big) \tag{39}$$

$$= \mathbb{E}_{q_\lambda(\boldsymbol{f}_d)} \left[ \log \mathcal{N}(\boldsymbol{y}_d; \boldsymbol{f}_d(\boldsymbol{X}), \sigma_d^2) \right] - \mathrm{KL}\big(q_\lambda(\boldsymbol{u}_d)||p(\boldsymbol{u}_d|\boldsymbol{Z}_d)\big), \tag{40}$$

where we defined $q_\lambda(\boldsymbol{f}_d) = \int p(\boldsymbol{f}_d|\boldsymbol{u}_d) q_\lambda(\boldsymbol{u}_d) d\boldsymbol{u}_d$ with the Euclidean variational distribution $q_\lambda(\boldsymbol{u}_d) = \mathcal{N}(\boldsymbol{u}_d; \tilde{\boldsymbol{\mu}}_d, \tilde{\boldsymbol{\Sigma}}_d)$, and wrote $p(\boldsymbol{u}_d|\boldsymbol{Z}_d) = p(\boldsymbol{u}_d)$ for simplicity. The inequality Eq. 35 corresponds to Jensen's inequality, while Eq. 38 is shown in (Titsias, 2009).

Finally, substituting Eq. 40 in Eq. 31 results in the following bound on the marginal likelihood

$$\log p(\boldsymbol{Y}) \geq \sum_{n=1}^{N} \sum_{d=1}^{D} \mathbb{E}_{q_\phi(\boldsymbol{x}_n)} \left[ \mathbb{E}_{q_\lambda(f_{n,d})} \left[ \log \mathcal{N}(y_{n,d}; f_{n,d}(\boldsymbol{x}_n), \sigma_d^2) \right] \right]$$

$$- \sum_{d=1}^{D} \mathrm{KL}\big(q_\lambda(\boldsymbol{u}_d)||p(\boldsymbol{u}_d|\boldsymbol{Z}_d)\big) - \sum_{n=1}^{N} \mathrm{KL}\big(q_\phi(\boldsymbol{x}_n)||p(\boldsymbol{x}_n)\big). \tag{41}$$

## D   MATÉRN KERNELS ON TAXONOMY GRAPHS

As explained in § 4 of the main paper, we leverage the Matérn kernel on graphs proposed by Borovitskiy et al. (2021) to design a kernel for our back-constrained GPHLVM that accounts for the geometry of the taxonomy graph. Here we provide the main equations of such a kernel, and refer the reader to (Borovitskiy et al., 2021) for further details. Formally, let us define a graph $G = (V, E)$ with vertices $V$ and edges $E$ and the *graph Laplacian* as $\boldsymbol{\Delta} = \boldsymbol{D} - \boldsymbol{W}$, where $\boldsymbol{W}$ is the graph adjacency matrix and $\boldsymbol{D}$ its corresponding diagonal degree matrix, with $\boldsymbol{D}_{ii} = \sum_j \boldsymbol{W}_{ij}$. The eigendecomposition $\boldsymbol{U}\boldsymbol{\Lambda}\boldsymbol{U}^\mathsf{T}$ of the Laplacian $\boldsymbol{\Delta}$ is then used to formulate both the SE and Matérn kernels on graphs, as follows,

$$k_{\infty,\kappa}^{\mathbb{G}}(c_n, c_m) = \boldsymbol{U} \left( e^{-\frac{\kappa^2}{2}\boldsymbol{\Lambda}} \right) \boldsymbol{U}^\mathsf{T}, \quad \text{and} \quad k_{\nu,\kappa}^{\mathbb{G}}(c_n, c_m) = \boldsymbol{U} \left( \frac{2\nu}{\kappa^2} + \boldsymbol{\Lambda} \right)^{-\nu} \boldsymbol{U}^\mathsf{T}, \tag{42}$$

where $\kappa$ is the lengthscale (i.e., it controls how distances are measured) and $\nu$ is the smoothness parameter determining mean-squared differentiability of the associated Gaussian process (GP). Note that the graph kernel expressions in Eq. 42 are obtained by considering the connection between Matérn kernel GPs and stochastic partial differential equations, originally proposed by Whittle (1963) and later extended to Riemannian manifolds in (Borovitskiy et al., 2020). This connection establishes that SE and Matérn GPs satisfy

$$e^{-\frac{\kappa^2}{4}\boldsymbol{\Delta}} \boldsymbol{f} = \mathcal{W}, \quad \text{and} \quad \left( \frac{2\nu}{\kappa^2} + \boldsymbol{\Delta} \right)^{\frac{\nu}{2}} \boldsymbol{f} = \mathcal{W}, \tag{43}$$

where $\mathcal{W} \sim \mathcal{N}(\boldsymbol{0}, \boldsymbol{I})$ and $\boldsymbol{f}: V \to \mathbb{R}$, which lead to definition of graph GPs (Borovitskiy et al., 2021).

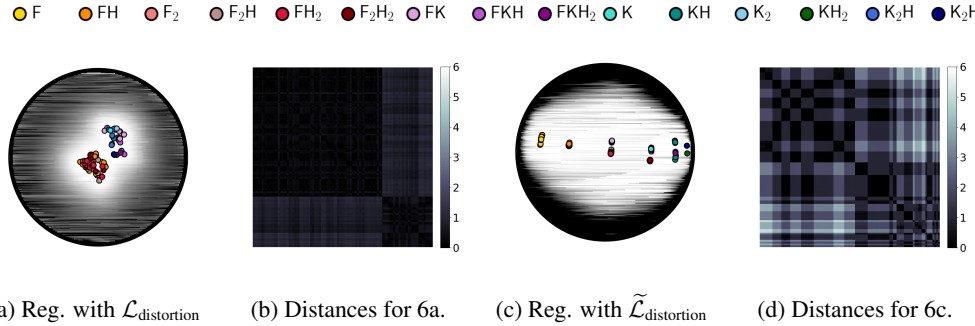

(a) Reg. with $\mathcal{L}_{\text{distortion}}$     (b) Distances for 6a.     (c) Reg. with $\widetilde{\mathcal{L}}_{\text{distortion}}$     (d) Distances for 6c.

Figure 6: Embeddings learned with distortion regularization. *(a)* and *(c)* display the latent embeddings after training our GPHLVM model with an added distortion loss $\mathcal{L}_{\text{distortion}}$ as it was originally defined, and with our modified distortion loss $\widetilde{\mathcal{L}}_{\text{distortion}}$, respectively. These embeddings indeed show that our regularizations failed to encode the distances in the graph (comparing the distances provided in *(b)* and *(d)* with Fig. 3).

## E    DISTORTION LOSS

As explained in the paper, we focus on two ways of embedding the graph in the hyperbolic space: a global approach using a stress regularization which matches graph distances with geodesic distances, and a combination between this stress regularization and the use of back constraints (see § 4). However, the literature on graph embeddings also surveys a *distortion loss* (Cruceru et al., 2021) given by

$$\mathcal{L}_{\text{distortion}}(\boldsymbol{X}) = \sum_{i<j} \left| \frac{\text{dist}_{\mathbb{H}^Q}(\boldsymbol{x}_i, \boldsymbol{x}_j)^2}{\text{dist}_{\mathbb{G}}(c_i, c_j)^2} - 1 \right|^2, \tag{44}$$

which tries to match the graph and manifold distances by minimizing their ratio's distance to 1.

We found that our problem is more subtle than usual graph embeddings, given that several points in our dataset may correspond to the same graph node (e.g., two different poses in which the left foot is the only limb in contact). Indeed, notice that Eq. 45 is ill-defined for the case $i = j$ (or equivalently $\text{dist}_{\mathbb{G}}(c_i, c_j)^2 = 0$). This is because all nodes $\boldsymbol{x}_i$ are assumed to be different from each other. However, in our setup, several $\boldsymbol{x}_i$ may correspond to the exact same class in the taxonomy.

Our first attempt to remediate this was to add a simple regularizer $\varepsilon = 10^{-1}$ to the denominator. However, this caused the loss to give more weight to the points where $\text{dist}_{\mathbb{G}}(c_i, c_j)^2 = 0$ (see Fig. 6a-6b for the outcome of training a GPHLVM with this type of regularization). We then considered an alternate definition of distortion in which the term inside the sum is given by

$$\widetilde{\mathcal{L}}_{\text{distortion}}(\boldsymbol{x}_i, \boldsymbol{x}_j) = \begin{cases} \lambda_1 \, \text{dist}_{\mathbb{H}^Q}(\boldsymbol{x}_i, \boldsymbol{x}_j) & \text{if } \boldsymbol{x}_i \text{ and } \boldsymbol{x}_j\text{'s classes are identical} \\ \lambda_2 \mathcal{L}_{\text{distortion}}(\boldsymbol{x}_i, \boldsymbol{x}_j) & \text{otherwise} \end{cases} \tag{45}$$

where $\lambda_1, \lambda_2 \in \mathbb{R}^+$ are hyperparameters. $\lambda_1$ governs how much we encourage latent codes of the same class to collapse into a single point, while $\lambda_2$ weights how much the geodesic distance should match the graph distance. After manual hyperparameter tuning, we obtained the latent space and distance matrix portrayed in Figs. 6c-6d. As can be seen in both accounts, the distortion loss produced lackluster results and failed to properly match the latent space distances with that of the graph. For these experiments, we used a loss scale of 50, $\lambda_1 = 0.01$ and $\lambda_2 = 10$, meaning that we strongly encouraged the distances between non-identical classes to match in ratio.

## F    ADDITIONAL DETAILS ON THE EXPERIMENTS OF § 5

### F.1    DATA

Table 3 describes the data of the whole-body support pose taxonomy used in the experiments reported in § 5. Each pose is identified with a support pose category, i.e., a node of the graph in Fig. 1-*right*, and with a set of associated contacts. As shown in the table, some support poses include several sets of contacts. For example, the support pose F groups all types of support poses

where only one foot is in contact with the environment. Notice that some sets of contacts are not represented in the data and thus do not appear in Table 3.

| Support pose | Contacts | Number |
|---|---|---|
| F | Left foot | 7 |
| | Right foot | 6 |
| FH | Left foot, left hand | 5 |
| | Right foot, right hand | 6 |
| | Left foot, right hand | 5 |
| | Right foot, left hand | 6 |
| $F_2$ | Left foot, right foot | 6 |
| $FH_2$ | Left foot, left hand, right hand | 6 |
| | Right foot, left hand, right hand | 6 |
| $F_2H$ | Left foot, right foot, left hand | 5 |
| | Left foot, right foot, right hand | 7 |
| $F_2H_2$ | Left foot, right foot, left hand, right hand | 7 |
| K | Left knee | 1 |
| | Right knee | 1 |
| FK | Left foot, right knee | 2 |
| | Right foot, left knee | 3 |
| KH | Left knee, left hand | 4 |
| | Right knee, right hand | 1 |
| $K_2$ | Left knee, right knee | 1 |
| FKH | Right foot, left knee, left hand | 5 |
| | Left foot, right knee, right hand | 2 |
| $KH_2$ | Left knee, left hand, right hand | 1 |
| $K_2H$ | Left knee, right knee, left hand | 2 |
| | Left knee, right knee, right hand | 1 |
| $FKH_2$ | Right foot, left knee, left hand, right hand | 2 |
| $K_2H_2$ | Left knee, right knee, left hand, right hand | 2 |

Table 3: Poses description extracted from the whole-body support pose taxonomy (Borràs et al., 2017) used in § 5 and App. G.

### F.2 TRAINING PARAMETERS AND PRIORS

Table 4 describes the hyperparameters used for the experiments reported in § 5 and App. G. We used the hyperbolic kernels defined in § 3.1 for the GPHLVMs, and the classical SE kernel for the Euclidean models. For the back-constraints mapping Eq. 11, we defined $k^{\mathbb{R}^D}(\boldsymbol{y}_n, \boldsymbol{y}_m)$ as the product of a Euclidean SE kernel with lengthscale $\kappa_{\mathbb{R}^D}$, and $k^{\mathbb{G}}(c_n, c_m)$ as a graph Matérn kernel with smoothness $\nu = 2.5$ and lengthscale $\kappa_{\mathbb{G}}$. We additionally scaled the product of kernels with a variance $\sigma_{\mathbb{R}^D, \mathbb{G}}$. For training the back-constrained GPHLVM and GPLVM, we used a Gamma prior $\text{Gamma}(\alpha, \beta)$ with shape $\alpha$ and rate $\beta$ on the lengthscale $\kappa$ of the kernels. The embeddings of the Euclidean models were initialized with PCA. For the GPHLVMs, the initial embeddings $\tilde{\boldsymbol{v}}$ obtained via PCA were transformed to elements of the tangent space $\mathcal{T}_{\boldsymbol{\mu}_0}\mathbb{H}^Q$ at the origin $\boldsymbol{\mu}_0$ by setting $\boldsymbol{v} = (0, \tilde{\boldsymbol{v}})^\top$ and then projected to the hyperbolic manifold using the exponential map. All models were trained by maximizing the loss $\mathcal{L} = \mathcal{L}_{\text{MAP}} - \gamma \mathcal{L}_{\text{stress}}$, where $\mathcal{L}_{\text{MAP}}$ denotes the log posterior of the model, $\mathcal{L}_{\text{stress}}$ is the stress-based regularization loss defined in Eq. 10, and $\gamma$ is a parameter balancing the two losses. The optimization was conducted using the Riemannian Adam optimizer (Bécigneul & Ganea, 2019) implemented in Geoopt (Kochurov et al., 2020) with a learning rate of $0.05$.

For the first part of the experiments on taxonomy expansion, we encoded unseen poses of each class for the back-constrained GPLVM and GPHLVM with a stress regularization using the models presented in Table 4. For the second part of the experiments, we left the class FH out during training and we "embedded" it using the back-constraints mapping. The newly-trained models also followed the same hyperparameters presented in Table 4.

| Experiment | Model | Regularization | Loss scale $\gamma$ | Prior on $\kappa_{\mathbb{H}/\mathbb{R}^Q}$ | $\kappa_{\mathbb{R}^D}$ | $\kappa_{\mathbb{G}}$ | $\sigma_{\mathbb{R}^D,\mathbb{G}}$ |
|---|---|---|---|---|---|---|---|
| Hyperbolic embeddings of support poses (§ 5) | GPLVM on $\mathbb{R}^2$ | No regularizer | 0 | None | - | - | - |
| | | Stress | 6 | None | - | - | - |
| | | BC + Stress | 1.3 | Gamma$(2,2)$ | 0.9 | 0.6 | 2 |
| | GPHLVM on $\mathbb{H}^2$ | No regularizer | 0 | None | - | - | - |
| | | Stress | 6 | None | - | - | - |
| | | BC + Stress | 1.3 | Gamma$(2,2)$ | 0.9 | 0.6 | 2 |
| Hyperbolic embeddings in $\mathbb{H}^3$ (App. G.1) | GPLVM on $\mathbb{R}^3$ | No regularizer | 0 | None | - | - | - |
| | | Stress | 10 | None | - | - | - |
| | | BC + Stress | 1.5 | Gamma$(2,2)$ | 0.9 | 0.6 | 2 |
| | GPHLVM on $\mathbb{H}^3$ | No regularizer | 0 | None | - | - | |
| | | Stress | 10 | None | - | - | - |
| | | BC + Stress | 1.5 | Gamma$(2,2)$ | 0.9 | 0.6 | 2 |
| Hyperbolic embeddings of standing poses (App. G.2) | GPLVM on $\mathbb{R}^2$ | No regularizer | 0 | None | - | - | - |
| | | Stress | 5 | None | - | - | - |
| | | BC + Stress | 0.7 | Gamma$(2,2)$ | 0.9 | 0.6 | 2 |
| | GPHLVM on $\mathbb{H}^2$ | No regularizer | 0 | None | - | - | - |
| | | Stress | 5 | None | - | - | - |
| | | BC + Stress | 0.7 | Gamma$(2,2)$ | 0.9 | 0.6 | 2 |
| Hyperbolic embeddings of standing poses with an augmented taxonomy (App. G.3) | GPLVM on $\mathbb{R}^3$ | No regularizer | 0 | None | - | - | - |
| | | Stress | 5 | None | - | - | - |
| | | BC + Stress | 1.5 | Gamma$(2,2)$ | 2.0 | 0.8 | 2 |
| | GPHLVM on $\mathbb{H}^3$ | No regularizer | 0 | None | - | - | - |
| | | Stress | 5 | None | - | - | - |
| | | BC + Stress | 1.5 | Gamma$(2,2)$ | 2.0 | 0.8 | 2 |

Table 4: Summary of experiments and list of hyperparameters.

### F.3 Marginal log-likelihoods of trained models

Table 5 shows the marginal log-likelihood (MLL) of the GPHLVM and GPLVM described in § 5. We observe that the hyperbolic models achieve a higher likelihood that their Euclidean counterparts.

| | Regularization | MLL | | Regularization | MLL |
|---|---|---|---|---|---|
| $\mathbb{R}^2$ | No reg. | -27.57 | $\mathbb{H}^2$ | No reg. | **-24.37** |
| | Stress | -55.33 | | Stress | **-49.20** |
| | BC+Stress | 5.43 | | BC+Stress | **7.67** |

Table 5: Marginal log-likelihood per geometry and regularization.

### F.4 Further details on trajectory generation via geodesics

Table 6 describes the transitions between support poses obtained by following the geodesic trajectories of the back-constrained GPHLVM and GPLVM with stress prior depicted in Fig. 2c. In contrast to GPHLVM, the Euclidean GPLVM often results in transitions that do not exist in the taxonomy. Interestingly, it also often uses more transitions than those originally needed. Notice that similar results are observed for the GPHLVM and GPLVM with stress prior depicted in Fig. 2b.

| Start | End | Transitions in $\mathbb{H}^2$ | Transitions in $\mathbb{R}^2$ |
|---|---|---|---|
| F | F$_2$H | F $\to$ FH $\to$ F$_2$H | F$\to$FH$_2$ $\to$ FH $\to$ F$_2$H |
| F | F$_2$H$_2$ | F $\to$ FH $\to$ F$_2$H $\to$ F$_2$H$_2$ | F$\to$FH$_2$ $\to$ F$_2$H$_2$ |
| F | FH$_2$ | F $\to$ FH $\to$ FH$_2$ | F$\to$FH$_2$ |
| F$_2$H | FH$_2$ | F$_2$H $\to$ FH $\to$ FH$_2$ | F$_2$H $\to$ FH $\to$ FH$_2$ |
| F | FK | F $\to$ F$_2$ $\to$ FK | F$\to$FH$_2$ $\to$ FH $\to$ F$_2$H $\to$ F$_2$$\to$FKH $\to$ FK |
| F$_2$ | K$_2$ | F$_2$ $\to$ FK $\to$ K$_2$ | F$_2$$\to$FKH $\to$ FK $\to$ FKH $\to$ FKH$_2$ $\to$ KH$_2$$\to$K$_2$ |
| FH | K$_2$H | FH $\to$ F$_2$H$\to$FH$_2$$\to$FKH $\to$ KH $\to$ K$_2$H | FH $\to$ F$_2$H $\to$ F$_2$$\to$FKH $\to$ FK $\to$ FKH $\to$ FKH$_2$$\to$K$_2$H |

Table 6: Transitions ($\to$) between classes of the taxonomy obtained by following the geodesic trajectories depicted in Fig. 2c. The classes and transitions correspond to the colors along the trajectories and match the class corresponding to the closest embedding at each point along the geodesic. Transitions that do not exist in the taxonomy are denoted as $\to$.

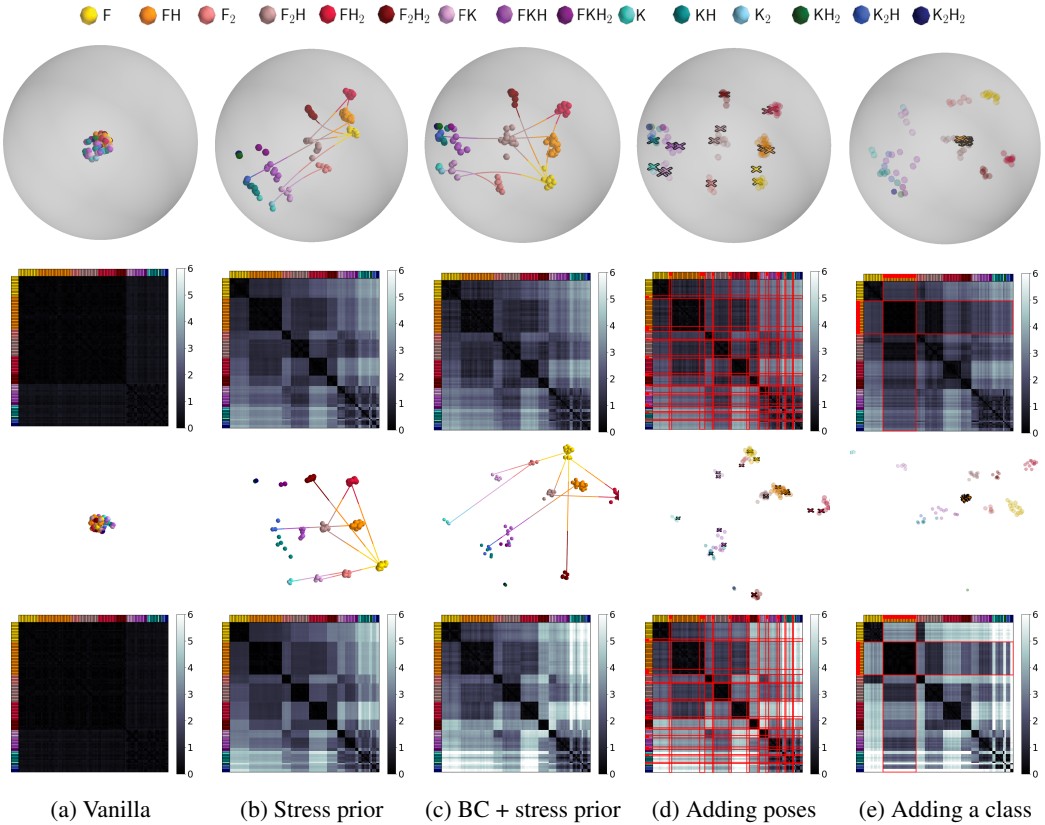

Figure 7: The first and last two rows respectively show the latent embeddings and examples of interpolating geodesics in $\mathcal{P}^3$ and $\mathbb{R}^3$, followed by pairwise distance matrices. Embeddings colors match those of Fig. 1-*right*. Added poses *(d)* and classes *(e)* are marked with crosses and highlighted with red in the distance matrices.

## G    ADDITIONAL EXPERIMENTS

### G.1    HYPERBOLIC EMBEDDINGS OF SUPPORT POSES IN $\mathbb{H}^3$

In this section, we embed the 100 poses used in § 5 into 3-dimensional hyperbolic and Euclidean spaces to analyze the performance of the proposed models in higher-dimensional latent spaces. Namely, we test the GPHLVM and GPLVM without regularization, with stress prior, and with back-constraints coupled with stress prior, similarly to the experiments on 2-dimensional latent spaces reported in the paper. Figs. 7a-7c show the learned embeddings alongside the corresponding distance matrices, which are to be compared with the graph distances in Fig. 3. As expected, and similarly to the 2-dimensional embeddings of Fig. 2a, the models without regularization do not encode any meaningful distance structure in the latent spaces (see Fig. 7a). In contrast, the models with stress prior result in embeddings that comply with the taxonomy graph structure, and the back constraints further organize the embeddings inside a class according to the similarity between their observations (see Figs. 7b-7c).

|  | Regularization | Stress $\pm\sigma$ |
|---|---|---|
| $\mathbb{R}^3$ | No regularizer | 4.04±4.38 |
|  | Stress | 0.18±0.34 |
|  | BC+Stress | 1.59±1.99 |
|  | — ” —: unseen poses | **0.16**±0.25 |
|  | — ” —: unseen class | 0.99±0.74 |
| $\mathbb{H}^3$ | No regularizer | 3.83±4.17 |
|  | Stress | **0.15**±0.22 |
|  | BC+Stress | **0.19**±0.28 |
|  | — ” —: unseen poses | 0.18±0.26 |
|  | — ” —: unseen class | **0.68**±0.74 |

Table 7: Average stress per geometry and regularization.

We observed a prominent stress reduction for the Euclidean 3-dimensional latent spaces compared to the 2-dimensional ones (see Table 7), as well as a reduction of non-existing transitions when following geodesic trajectories (see Table 8). This is due to the increase of volume available to match the graph structure in $\mathbb{R}^3$ relatively to $\mathbb{R}^2$. However, all Euclidean models are still

outperformed by the 2-dimensional hyperbolic embeddings presented in § 5 (see Table 1). This is due to the fact that the volume of balls in hyperbolic space increases exponentially with respect to the radius of the ball rather than polynomially as in Euclidean space. In other words, the geometry of the hyperbolic manifold increases the volume available to match the graph structure compared to Euclidean spaces, thus resulting in better low-dimensional representations of taxonomy data. Notice that the GPHLVM models with 3-dimensional hyperbolic latent space result in a similar or slightly reduced stress compared to their 2-dimensional counterparts (presented in § 5). This indicates that the volume of the 2-dimensional hyperbolic latent space is sufficient to represent the considered data. Moreover, similarly as for the 2-dimensional cases, the back-constrained GPHLVM and GPLVM allow us to properly place unseen poses or taxonomy classes into the latent space (see Figs. 7d-7e).

| Start | End | Transitions in $\mathbb{H}^3$ | Transitions in $\mathbb{R}^3$ |
|---|---|---|---|
| F | $F_2H$ | $F \rightarrow FH \rightarrow F_2H$ | $F \rightarrow FH \rightarrow F_2H$ |
| F | $F_2H_2$ | $F \rightarrow FH \rightarrow F_2H \rightarrow F_2H_2$ | $F \rightarrow FH \textcolor{red}{\rightarrow} F_2H_2$ |
| F | $FH_2$ | $F \rightarrow FH \rightarrow FH_2$ | $F \rightarrow FH \rightarrow FH_2$ |
| $F_2H$ | $FH_2$ | $F_2H \rightarrow FH \rightarrow FH_2$ | $F_2H \rightarrow FH \rightarrow FH_2$ |
| F | FK | $F \rightarrow F_2 \rightarrow FK$ | $F \rightarrow F_2 \rightarrow FK$ |
| $F_2$ | $K_2$ | $F_2 \rightarrow FK \rightarrow K_2$ | $F_2 \rightarrow FK \rightarrow K_2$ |
| FH | $K_2H$ | $FH \rightarrow F_2H \rightarrow FKH \rightarrow KH \rightarrow K_2H$ | $FH \rightarrow F_2H \rightarrow FKH \rightarrow FKH_2 \textcolor{red}{\rightarrow} K_2H$ |

Table 8: Transitions ($\rightarrow$) between classes of the taxonomy obtained by following the geodesic trajectories depicted in Fig. 7c. The classes and transitions correspond to the colors along the trajectories and match the class corresponding to the closest embedding at each point along the geodesic. Transitions that do not exist in the taxonomy are denoted as $\textcolor{red}{\rightarrow}$.

### G.2 HYPERBOLIC EMBEDDINGS OF STANDING POSES

In this section, we consider a different subset of the whole-body support pose taxonomy, leading to a different graph. Namely, we use 60 standing poses of the dataset in (Mandery et al., 2016) and (Langenstein, 2020), which correspond to graph nodes of standing support poses (left side of the graph in Fig. 1). Specifically, we use a balanced dataset composed of 5 poses for each of the contact sets of the standing support poses described in Table 3. We embed the 60 poses into 2-dimensional hyperbolic and Euclidean spaces using GPHLVM and GPLVM. For each approach, we test the model without regularization, with stress prior, and with back-constraints coupled with stress prior using the parameters described in App. F.2 and Table 4.

Figs. 8a-8c show the learned embeddings alongside their corresponding distance matrices, which are to be compared with the graph distances in Fig. 9a. As for the previous experiments, the models with stress prior result in embeddings that comply with the taxonomy graph structure, with additional intra-class organizations for the back-constrained models. It is worth noticing that, despite the fact that the considered taxonomy graph is smaller than for the previous experiments, all Euclidean GPLVMs remain outperformed by the hyperbolic models, which better match the taxonomy structure (see also Table 9b). Similarly to the experiments reported in § 5, the back-constrained GPHLVM and GPLVM allow us to properly place unseen poses or taxonomy classes into the latent space (see Figs. 8d-8e). As mentioned in the main text, our GPHLVM intrinsically provides a mechanism to plan motions via geodesics in the low-dimensional latent space. Examples of geodesics between two standing poses are shown in Figs. 8b-8c, where the trajectory color matches the class corresponding to the closest latent point. The transitions between standing support poses obtained by following these geodesic trajectories are also described in Table 9. As for our previous experiments, the geodesics, i.e., shortest paths, in the GPHLVM latent space correspond to shortest paths in the taxonomy graph. Due to the size of the taxonomy graph, we observe fewer forbidden (i.e. nonexistent) transitions than for the previous experiments in the Euclidean models. However, as their latent space does not match the taxonomy structure, they often require additional transitions and thus do not follow shortest paths in the taxonomy graph.

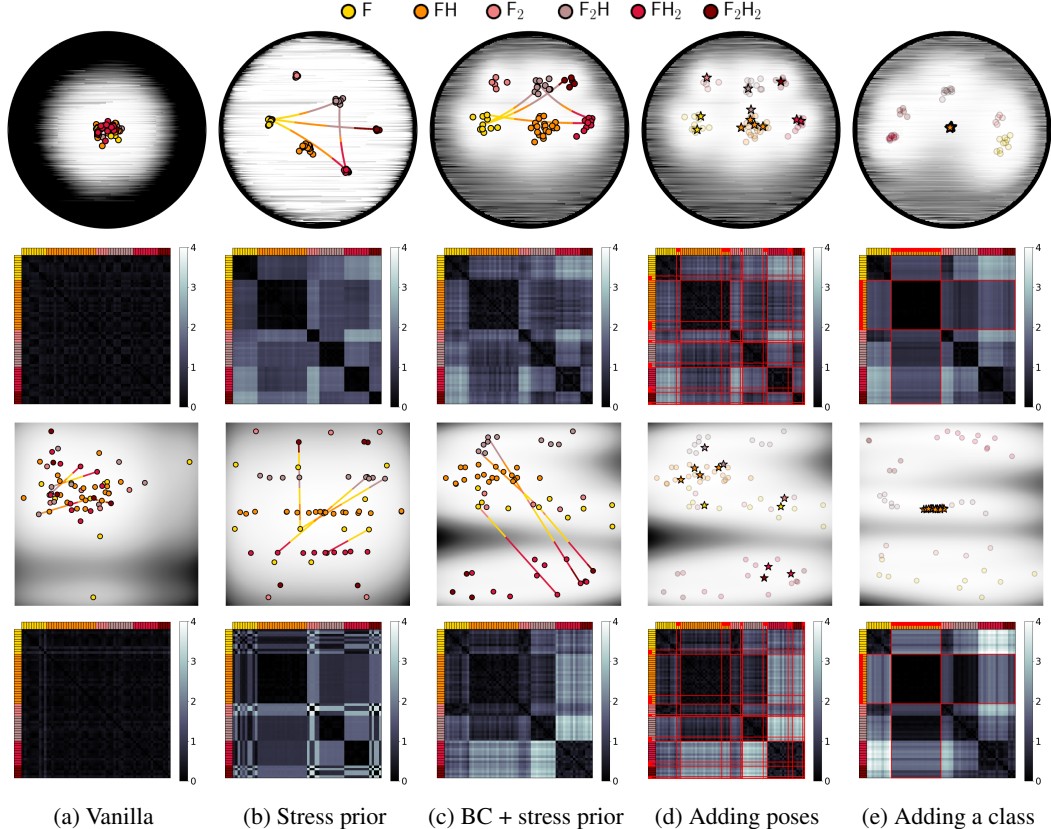

| (a) Vanilla | (b) Stress prior | (c) BC + stress prior | (d) Adding poses | (e) Adding a class |

Figure 8: Embeddings of standing poses: The first and last two rows respectively show the latent embeddings of bipedal poses and examples of interpolating geodesics in $\mathcal{P}^2$ and $\mathbb{R}^2$, followed by pairwise distance matrices. Embeddings colors match those of Fig. 1-*right*, and background colors indicate the GPLVM uncertainty. Added poses *(d)* and classes *(e)* are marked with stars and highlighted with red in the distance matrices.

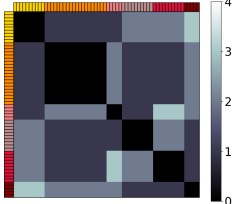

| | Regularization | Stress $\pm\sigma$ |
|---|---|---|
| $\mathbb{R}^2$ | No regularizer | 1.69±1.96 |
| | Stress | 0.62±1.41 |
| | BC+Stress | 0.68±0.96 |
| | — ” —: unseen poses | 0.61±0.84 |
| | — ” —: unseen class | 0.47±0.38 |
| $\mathbb{H}^2$ | No regularizer | 1.66±1.95 |
| | Stress | **0.07**±0.09 |
| | BC+Stress | **0.15**±0.18 |
| | — ” —: unseen poses | **0.17**±0.20 |
| | — ” —: unseen class | **0.22**±0.31 |

| (a) Graph distance between the standing poses. | (b) Average stress per geometry and regularization. |

Figure 9: Embeddings of standing poses: *(a)* shows the graph distance following the left part of Fig. 1-*right*. *(b)* shows the stress resulting from the different embeddings of standing poses.

| Start | End | Transitions in $\mathbb{H}^2$ | Transitions in $\mathbb{R}^2$ |
|-------|-----|---------------------------------|--------------------------------|
| F | $F_2H$ | $F \to FH \to F_2H$ | $F \to F_2 \to F \to F_2 \to FH \to F_2H$ |
| F | $F_2H_2$ | $F \to FH \to F_2H \to F_2H_2$ | $F \to F_2 \to F \to FH_2 \to F_2H_2$ |
| F | $FH_2$ | $F \to FH \to FH_2$ | $F \to F_2 \to F \to FH_2$ |
| $F_2H$ | $FH_2$ | $F_2H \to FH \to FH_2$ | $F_2H \to FH \to F_2 \to F \to F_2 \to F \to FH_2 \to F_2H_2 \to FH_2$ |

Table 9: Embeddings of standing poses: Transitions ($\to$) between classes of the taxonomy obtained by following the geodesic trajectories depicted in Fig. 8c. The classes and transitions correspond to the colors along the trajectories and match the class corresponding to the closest embedding at each point along the geodesic. Transitions that do not exist in the taxonomy are denoted as $\to$.

### G.3 HYPERBOLIC EMBEDDINGS OF STANDING POSES WITH AN AUGMENTED TAXONOMY FOR IMPROVED TRAJECTORY GENERATION

As shown in § 5, geodesics in the hyperbolic latent space of our GPHLVM intrinsically provide a mechanism to plan motions accounting for the underlying taxonomy. However, as discussed in § 5, the whole-body support pose taxonomy (Borràs et al., 2017) lacks information about the type of contact in the considered poses, thus leading to artifacts in the geodesic-generated motions. In the main paper, we showed that the quality of the generated motion is improved by augmenting the whole-body support pose taxonomy with additional contact information. To do so, we considered an *augmented* whole-body support pose taxonomy which explicitly distinguishes between left and right contacts. In other words, the nodes and transitions of Fig. 1-*right* are adapted to consider left and right contacts. For instance, the 1-foot contact (F) node is separated into left-foot ($F^l$) and right-foot ($F^r$) contact nodes. To facilitate motion planning and to test the GPHLVM ability of dealing with high-dimensional spaces, we represent each pose as a vector $\boldsymbol{y}_n \in \mathbb{R}^{44}$ of joint angles instead of a vector of hands and feet positions.

We embed the 60 standing poses described in App. G.2 into 3-dimensional hyperbolic and Euclidean spaces using GPHLVM and GPLVM, respectively. For each approach, we test the model without regularization, with stress prior, and with back-constraints coupled with stress prior using the parameters described in App. F.2 and Table 4. Figs. 10a-10c show the learned embeddings alongside their corresponding distance matrices, which are to be compared with the graph distances of the augmented taxonomy in Fig. 11a. Similarly to previous experiments, the models with stress prior result in embeddings complying with the taxonomy graph structure (Fig. 10b), with additional intra-class organizations for the back-constrained models (Fig. 10c). Notice that the embeddings differentiate between left and right contacts according to the augmented taxonomy: For instance, we observe four clusters of orange embeddings corresponding to $F^lH^l$, $F^lH^r$, $F^rH^l$, and $F^rH^r$. As shown in Table 11b, the hyperbolic models better represent the taxonomy structure and outperform the Euclidean models. Similarly to previous experiments, the back-constraint mapping introduced in § 4 allows us to properly place unseen poses or taxonomy classes into the latent space (see Figs. 10d-10e).

Examples of motions planned by following geodesics between two standing poses in the hyperbolic latent space are displayed in the main paper (Fig. 5). The corresponding geodesics are shown in Fig. 10c, with the colors along the trajectory matching the class corresponding to the closest hyperbolic latent point. The resulting transitions are given in Table 10. As mentioned in the main paper, we observe that, in contrast to the trajectories of Fig. 4, the motions generated by considering the augmented taxonomy (Fig. 5) result in more realistic – human-like – interpolations between the given initial and final poses. Moreover, these motions look more realistic than the motions obtained via linear interpolation in the Euclidean latent space of the vanilla back-constrained GPLVM. As shown in Fig. 12, the motions planned in the Euclidean latent space sometimes result in unrealistic joint configurations and the same posture is associated with different types of contacts (see middle part of the motions). As shown in Table 10, non-existing transitions arise more frequently when following trajectories generated by the Euclidean model.

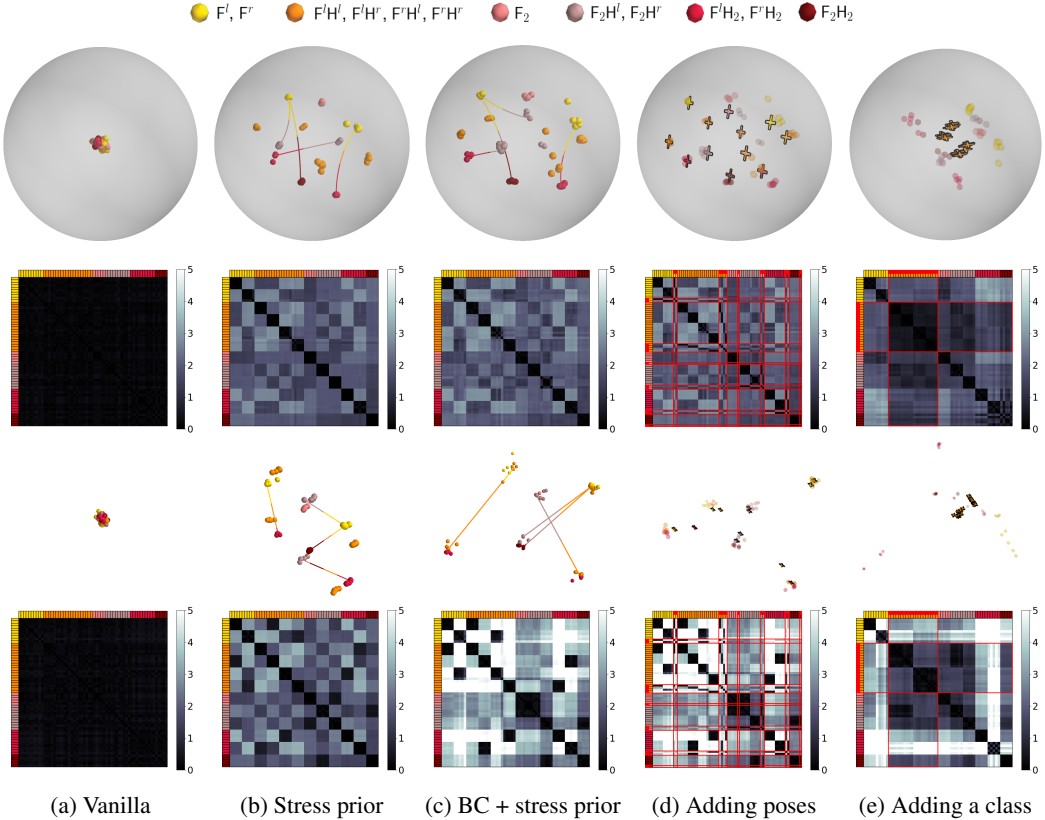

Figure 10: Embeddings of standing poses considering the augmented whole-body support pose taxonomy: The first and last two rows respectively show the latent embeddings and examples of interpolating geodesics in $\mathcal{P}^3$ and $\mathbb{R}^3$, followed by pairwise distance matrices. Embeddings colors match those of Fig. 1-*right*. Added poses *(d)* and classes *(e)* are marked with crosses and highlighted with red in the distance matrices.

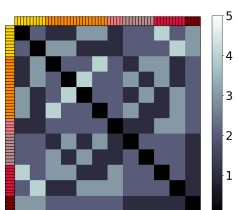

(a) Graph distance between the standing poses.

| | Regularization | Stress $\pm\sigma$ |
|---|---|---|
| $\mathbb{R}^3$ | No regularizer | 3.85±3.63 |
| | Stress | 0.31±0.31 |
| | BC+Stress | 4.92±7.60 |
| | — ” —: unseen poses | 1.93±2.34 |
| | — ” —: unseen class | **2.31**±3.24 |
| $\mathbb{H}^3$ | No regularizer | 4.05±3.77 |
| | Stress | **0.23**±0.34 |
| | BC+Stress | **1.25**±2.14 |
| | — ” —: unseen poses | **1.60**±2.21 |
| | — ” —: unseen class | 3.20±4.59 |

(b) Average stress per geometry and regularization.

Figure 11: Embeddings of standing poses considering the augmented whole-body support pose taxonomy: *(a)* shows the graph distance (colors follow Fig. 1-*right*). *(b)* shows the stress resulting from the different embeddings of standing poses.

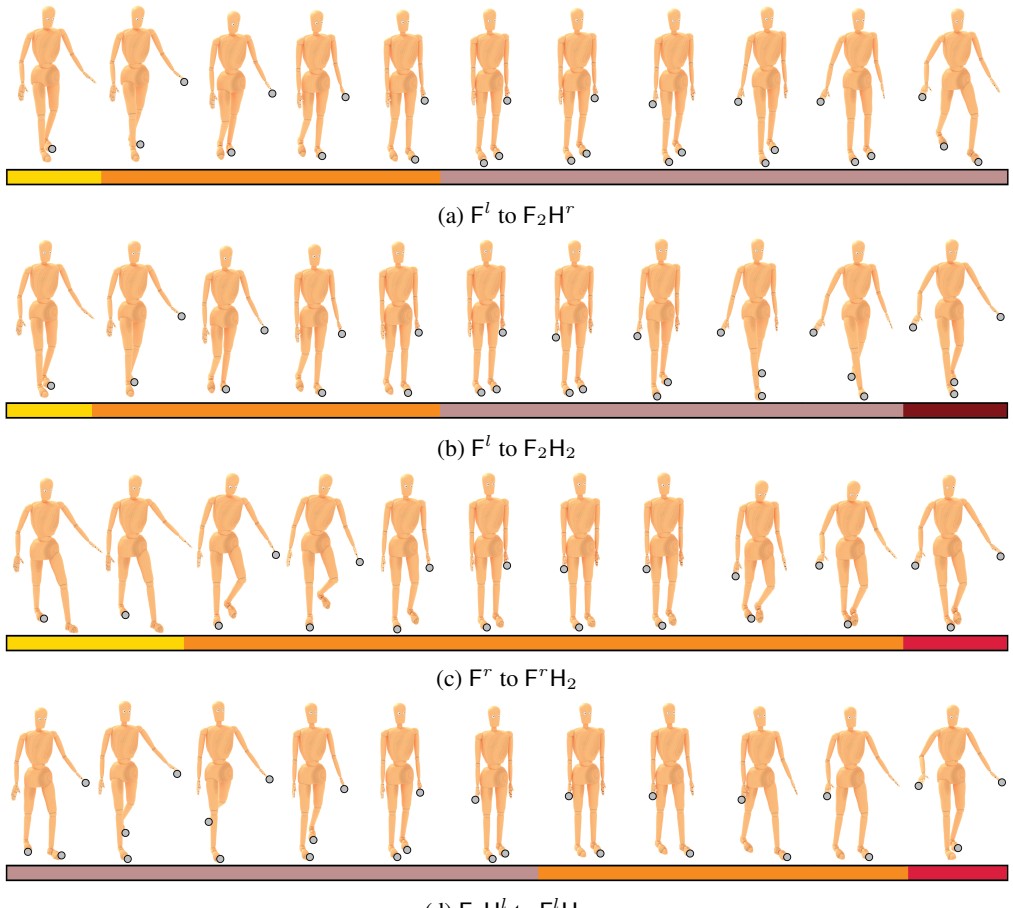

(a) $F^l$ to $F_2H^r$

(b) $F^l$ to $F_2H_2$

(c) $F^r$ to $F^rH_2$

(d) $F_2H^l$ to $F^lH_2$

Figure 12: Motions obtained via linear interpolation in the latent space of the vanilla Euclidean back-constrained GPHLVM trained on the augmented taxonomy (Fig. 10c). Contacts are denoted by gray circles. The colorbars identify the support pose of the closest pose in the latent space.

| Start | End | Transitions in $\mathbb{H}^3$ | Transitions in $\mathbb{R}^3$ |
|---|---|---|---|
| $F^l$ | $F_2H^r$ | $F^l \rightarrow F_2H^r$ | $F^l \rightarrow F^lH^l \rightarrow F_2H^l \rightarrow F_2H^r$ |
| $F^l$ | $F_2H_2$ | $F^l \rightarrow F_2H^l \rightarrow F_2H_2$ | $F^l \rightarrow F^lH^l \rightarrow F_2H^l \rightarrow F_2H^r \rightarrow F_2H_2$ |
| $F^r$ | $F^rH_2$ | $F^r \rightarrow F^rH^r \rightarrow F^rH_2$ | $F^r \rightarrow F^rH^l \rightarrow F^rH^r \rightarrow F^rH_2$ |
| $F_2H^l$ | $F^lH_2$ | $F_2H^l \rightarrow F_2H_2 \rightarrow F^lH_2$ | $F_2H^l \rightarrow F_2H^r \rightarrow F^lH^r \rightarrow F^lH_2$ |

Table 10: Embeddings of standing poses considering the augmented whole-body support pose taxonomy: Transitions ($\rightarrow$) between classes of the taxonomy obtained by following the geodesic trajectories depicted in Fig. 10. The classes and transitions correspond to the colors along the trajectories and match the class corresponding to the closest embedding at each point along the geodesic. Transitions that do not exist in the taxonomy are denoted as $\rightarrow$.

## G.4 COMPARISON AGAINST VARIATIONAL AUTOENCODERS

**Hyperbolic embeddings of support poses:** In this section, we compare the trained GPHLVMs of Fig. 2 with two additional baselines: a vanilla variational autoencoder (VAE) and a hyperbolic variant of this VAE in which the latent space is the Lorentz model of hyperbolic geometry (akin to Mathieu et al. (2019)). Both VAEs are designed with 12 input nodes, 6 hidden nodes, a 2-dimensional latent space, and a symmetric decoder. Their encoder specifies the mean and standard deviation of a normal distribution (resp. wrapped normal for the hyperbolic VAE), and their decoder specifies the mean and standard deviation of the normal distribution that governs the reconstructions. Both models are trained by maximizing an Evidence Lower Bound (ELBO) under similar regimes as

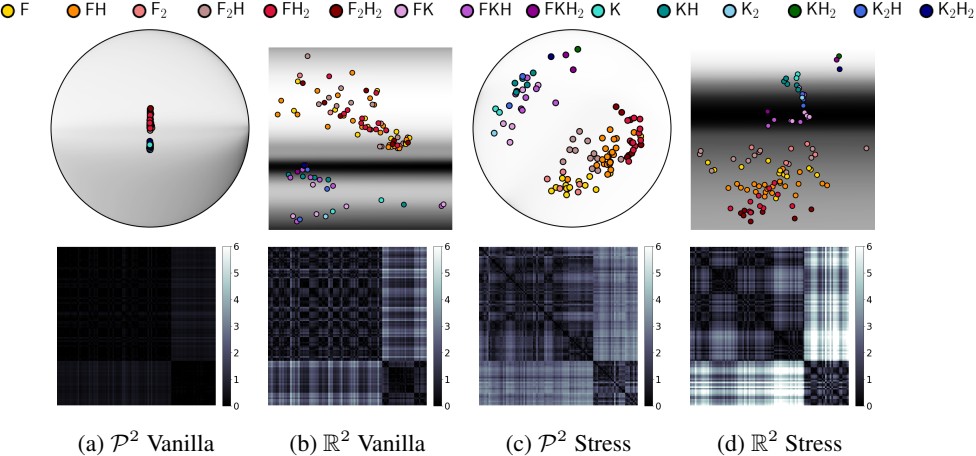

(a) $\mathcal{P}^2$ Vanilla     (b) $\mathbb{R}^2$ Vanilla     (c) $\mathcal{P}^2$ Stress     (d) $\mathbb{R}^2$ Stress

Figure 13: Embeddings of the VAE baselines: The first and second rows show the latent spaces of the (hyperbolic) VAE and the distance matrix between the latent codes, respectively. When comparing these distance matrices and encodings with that of our GPHLVMs (see Fig. 2), we notice that our proposed model is better able to preserve the graph distance structure. We argue this is because VAEs enforce latent spaces that follow a unit Gaussian, which is an opposite goal to ours.

the GPHLVMs, i.e., $1000$ epochs with a learning rate of $0.05$. The KL divergence for the hyperbolic VAE is computed using Monte Carlo estimates.

Importantly, the VAE models only seem to capture a global structure that separates standing from kneeling poses (except the vanilla hyperbolic VAE in Fig. 13a). Although adding a stress regularization with the same scale as for the GPHLVM ($\gamma = 6$) helps preserve the graph distance structure, the embeddings organization is still not competitive with the one achieved by our GPHLVM models (see Fig. 2). Moreover, when compared to our proposed GPHLVM, all VAE models provide a subpar uncertainty modeling in their latent spaces.

|  | Regularization | Stress $\pm\sigma$ |
|---|---|---|
| $\mathbb{R}^2$ | No reg. | 1.88±2.57 |
|  | Stress | 0.59±0.89 |
| $\mathbb{H}^2$ | No reg. | 3.96±4.22 |
|  | Stress | 0.52±0.71 |

Table 11: Average stress per geometry and regularization for VAE baselines.

Table 11 shows that the average stress of the latent embeddings for the VAE baselines (trained with and without stress regularization) is higher than the average stress of our models (see Table 1). Overall, our proposed GPHLVM consistently outperforms all VAEs to encode meaningful taxonomy information in the latent space. We argue that VAEs are not the right tool for our target applications. When training VAEs, the Kullback-Leibler term in the ELBO tries to regularize the latent space to match a unit Gaussian. This regularization is in stark contrast with our goal of separating the embeddings to preserve the taxonomy graph distances.

**Hyperbolic embeddings of standing poses with an augmented taxonomy:** We further compare our GPHLVM model against the vanilla and hyperbolic VAEs in the experiment described in Sec. G.3. Namely, we consider the *augmented* whole-body support pose taxonomy which explicitly distinguishes between left and right contacts and we represent each pose as a vector of joint angles. This increases the dimensionality of the data to 44.

We tested the vanilla and hyperbolic VAEs without regularization and with a stress regularization with the same scale as for the GPHLVM ($\gamma = 1.5$). Fig. 14 shows the learned embeddings alongside distance matrices, which are to be compared with the GPHLVM model of Figs. 10b-10c and with the ground-truth graph distances of Fig. 11a. Despite the stress regularization, the VAEs' tendency to have unit-normally distributed latent representations hinders the distance matching. This is further quantified by the mean stress presented in Table 12, which show a higher mean stress ($0.34$ and $0.44$) than our model in the same taxonomy ($0.23$, see Table 11b).

|  | Regularization | Stress $\pm\sigma$ |
|---|---|---|
| $\mathbb{R}^3$ | No reg. | 2.33±3.10 |
|  | Stress | 0.34±0.37 |
| $\mathbb{H}^3$ | No reg. | 3.01±3.13 |
|  | Stress | 0.44±0.63 |

Table 12: Average stress per geometry and regularization for VAE baselines trained on the augmented taxonomy (see App. G.3).

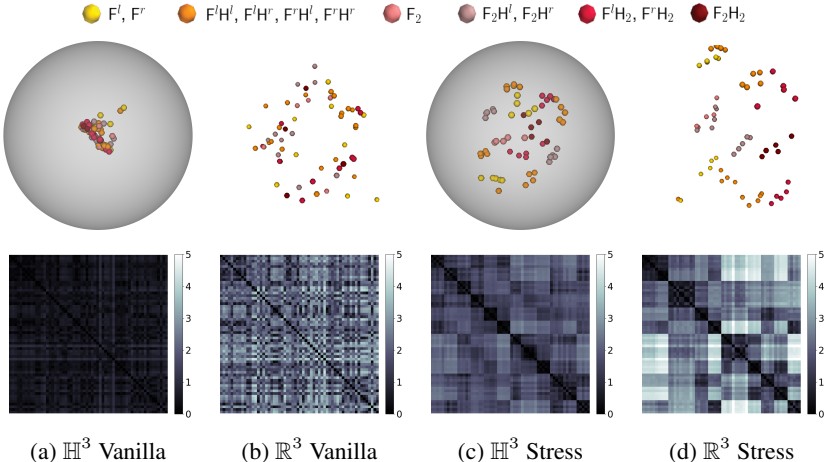

(a) $\mathbb{H}^3$ Vanilla     (b) $\mathbb{R}^3$ Vanilla     (c) $\mathbb{H}^3$ Stress     (d) $\mathbb{R}^3$ Stress

Figure 14: Embeddings of the VAE baselines considering the augmented whole-body support pose taxonomy: The first and second rows show the latent spaces of the (hyperbolic) VAE and the distance matrix between the latent codes, respectively.

| | **GPHLVM** $Q = 2$ | **GPHLVM** $Q = 3$ | **GPLVM** $Q = 2$ | **GPLVM** $Q = 3$ |
|---|---|---|---|---|
| Training | $2.5 \times 10^3$ | $8.91$ | $5.9$ | $6.3$ |
| Decoding | $1.33 \times 10^{-2}$ | $1.57 \times 10^{-5}$ | $1.16 \times 10^{-5}$ | $1.22 \times 10^{-5}$ |

Table 13: Average runtime (in seconds) for training and decoding phases of our GPHLVM and vanilla GPLVM over 5 experiments, using 2 and 3-dimensional latent spaces for both models. Training time was measured over 500 iterations for both models. The implementations are fully developed on Python, and runtime measurements were taken using a standard laptop with 32 GB RAM, Intel Xeon CPU E3-1505M v6 processor and Ubuntu 20.04 LTS.

Fig. 15 shows examples of motions planned by following geodesics between two standing poses in the hyperbolic VAE latent space. Similarly as the motions generated in the latent space of the proposed GPHLVM (Fig. 5), these motions result in realistic interpolations between the given initial and final poses.

## G.5   RUNTIME

In order to show the computational cost of our approach, we ran a set of experiments to measure the average runtime for the training and decoding phases, using 2 and 3-dimensional latent spaces. As a reference, we added the runtime measurements of Euclidean counterpart, that is, the vanilla GPLVM. Table 13 shows the runtime measurements. Note that the main computational burden arises in our GPLHVM with a 2-dimensional latent space, which is in sharp contrast with the experiments using a 3-dimensional latent space. As discussed in the main paper, this increase in computational cost is mainly attributed to the 2-dimensional hyperbolic kernel.

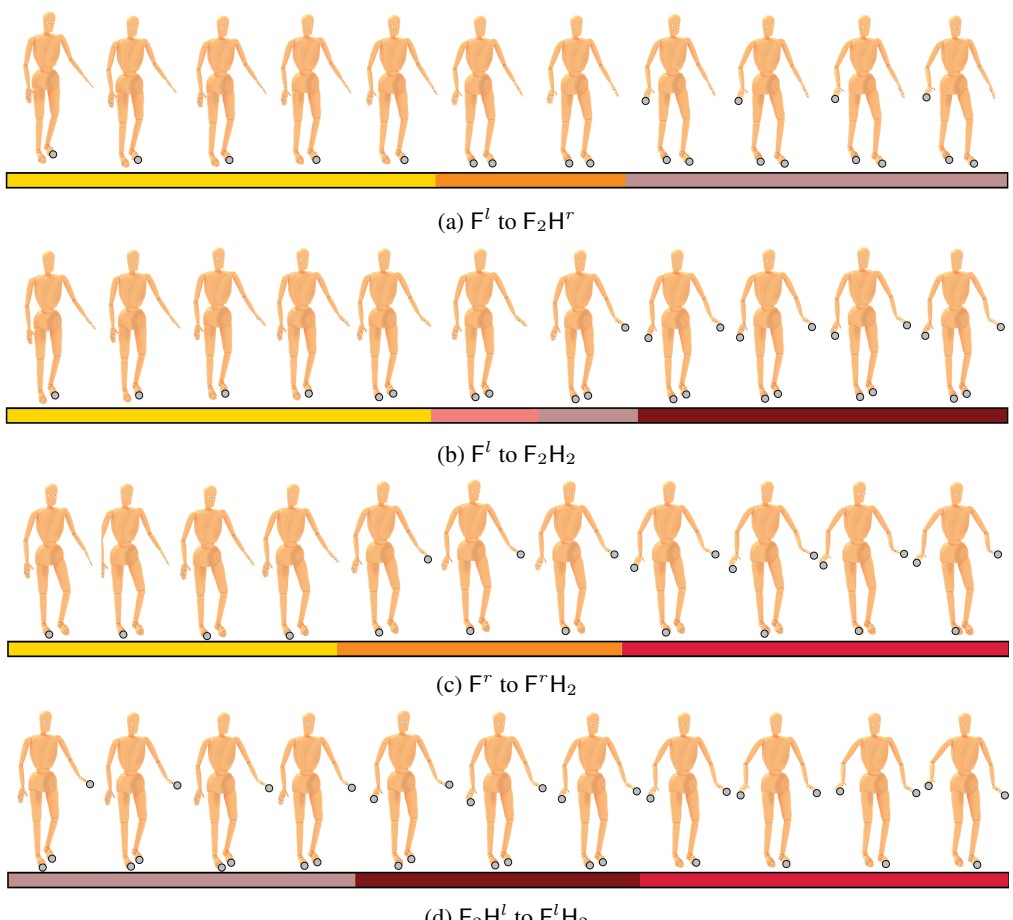

(a) $\mathsf{F}^l$ to $\mathsf{F}_2\mathsf{H}^r$

(b) $\mathsf{F}^l$ to $\mathsf{F}_2\mathsf{H}_2$

(c) $\mathsf{F}^r$ to $\mathsf{F}^r\mathsf{H}_2$

(d) $\mathsf{F}_2\mathsf{H}^l$ to $\mathsf{F}^l\mathsf{H}_2$

Figure 15: Motions obtained via geodesic interpolation in the latent space of the hyperbolic VAE trained on the augmented taxonomy (Fig. 14c). Contacts are denoted by gray circles. The colorbars identify the support pose of the closest pose in the latent space.

