# OpenReview forum: "Bringing robotics taxonomies to continuous domains via GPLVM on hyperbolic manifolds"
_ICLR.cc/2023/Conference — Submitted to ICLR 2023_

### Official Review · Reviewer_1Gzu · 2022-10-24

**Confidence:** 2
**Correctness:** 3
**Technical Novelty And Significance:** 3
**Empirical Novelty And Significance:** 2
**Recommendation:** 6

**Clarity, Quality, Novelty And Reproducibility:**

The paper is well-structured and written. For reproducibility, some hyperparameter and training details are provided in appendix but the code is not yet available.

### Minor Clarifications
- Is c_i used to denote the taxonomy node and the class? What are the added classes in Fig 2. (e) part?

**Strength And Weaknesses:**

### Strengths
The paper mentions the limitations and future directions for the community, such as the need to (1)  apply physical constraints on the generated motion, and (2) reduce the computational cost of the hyperbolic kernel. The lack of information on contact and object locations results in artifacts during interpolations between initial and final latent pose embeddings.

The paper is clearly written with sufficient additional details mentioned in the background and appendix sections.

Additional videos are provided for the generated trajectories on an augmented dataset with left & right side labels for hand and foot contacts.

### Weakness
It is unclear to understand the generated trajectories as the contacts are not visualized, only the human joints are shown.

**Summary Of The Paper:**

The paper proposes robot taxonomy, for hierarchical classifications of the human hand postures and body poses, through Gaussian process hyperbolic latent variable models (GPHLVM) built on top of existing GPLVM and using its graph-based distance-preserving priors and back constraints. The learned hyperbolic embeddings comply with the original graph structure and encode the unseen poses better than the euclidean ones. The assumption behind hyperbolic embeddings is that (1) the distance grows exponentially from the origin and (2) the shortest paths between 2 distant points tend to pass through the origin, thereby inducing a hierarchy. The model is evaluated on whole-body support pose taxonomy of 100 human poses of standing or kneeling and compared to euclidean pose embeddings. The ablation study shows that regularization with stress prior + back constraints improves clustering in Poincare ball P^2 and results in better latent space mapping of unseen poses and classes.

**Summary Of The Review:**

Overall, the paper provides a perspective on the benefits of using hyperbolic embeddings over euclidean ones for humanoid joint pose taxonomy. The training paradigm for learning embeddings for taxonomy is driven by existing work in terms of using graph-based priors and back constraints, with modifications for the proposed hyperbolic manifold formulation.
The two advantages highlighted for the learned taxonomy are (1) generalization to unseen poses and (2) interpolation for generating trajectories. The latter, however, suffers from a lack of representation of contacts due to self-collisions and obstacles - a noted limitation of the current work.

---

> ### Author Response · Authors · 2022-11-18
> **Response to comments of Reviewer 1Gzu**
>
> Thank you very much for your time reviewing our work! We are glad that the reviewer appreciated that we clearly stated the limitations of our work and that "sufficient additional details" are mentioned in appendices and in the background section.
>
> > *It is unclear to understand the generated trajectories as the contacts are not visualized, only the human joints are shown.*
>
> Thank you for this comment! We added the contact information in our video and updated it in the supplementary material.
>
> > Is $c_i$ used to denote the taxonomy node and the class? What are the added classes in Fig 2. (e) part?
>
> $c_i$ denotes the taxonomy node, or equivalently the class to which a given pose belong. In Fig. 2e, the class $F_1H_1$ was withhold during training and then added through the back-constrained mapping.
>
> **Reproducibility:** The code related to the paper will be made publicly available shortly.
>
> **Further clarifications:** If any particular issues in need of further clarification arise that were not apparent at the time of the review, we are happy to respond in follow-up comments as needed.

---

### Official Review · Reviewer_ttyr · 2022-10-25

**Confidence:** 1
**Correctness:** 4
**Technical Novelty And Significance:** 4
**Empirical Novelty And Significance:** 3
**Recommendation:** 8

**Clarity, Quality, Novelty And Reproducibility:**

* The writing is clear. The authors are thorough and provide extensive details in the appendix. The authors also generously provide background information (e.g. on GPLVMs, Riemannian geometry, the hyperbolic manifold, etc.)
* The work is also original, as it is the first to apply the hyperbolic manifold to learning robotics taxonomy representations to the best of my knowledge.

**Strength And Weaknesses:**

* The paper is well-written: the authors include extensive technical details in the main paper and appendix.
* The approach is novel: it is the first to leverage the hyperbolic manifold in encoding hierarchical structures for robotics applications.
* The experiments are sufficient in demonstrating the approach. The visualizations in Fig. 2 demonstrate that the embeddings comply with the graph structure, and that the hyperbolic approach outperforms the Euclidean counterpart even at higher dimensions.
* Though physical constraints are not yet considered, the motion generation results look promising and indicate that the learned representations work well.

**Summary Of The Paper:**

The authors propose to model robotics taxonomy data via a proposed Gaussian process hyperbolic latent variable model (GPHLVM). The GPHLVM maintains the structure of the human-designed taxonomy while embedding the taxonomy node features into a hyperbolic latent space, which naturally encodes a continuous hierarchical structure. The GPHLVM is an extension of GPLVM for hyperbolic spaces, and the authors propose using Riemannian optimization methods for training the GPHLVM and use graph-distance information as inductive bias to learn the embeddings, encouraging the embeddings to adhere to the taxonomy structure. They evaluate the approach on a whole-body support pose taxonomy dataset, and experiments show that the 2-dimensional encodings display desired structure complying with the taxonomy. Experiments also show trajectory generation by following paths in the latent space between two embeddings,

**Summary Of The Review:**

The authors introduce a novel idea of embedding robotics taxonomies into a hyperbolic manifold. This is a neat idea as the hyperbolic manifold is a natural representation for hierarchical structures such as trees and graphs. The paper is well written and includes extensive technical details and analysis, and the authors demonstrate nice applications on the whole-body pose dataset, showing various motions obtained via geodesic interpolation in the latent space.

---

> ### Author Response · Authors · 2022-11-18
> **Response to comments of Reviewer ttyr**
>
> Thank you very much for your review! We are delighted to read that our approach is novel, that our paper is "well-written" and that "the experiments are sufficient in demonstrating the approach"!
>
> If any particular issues in need of further clarification arise that were not apparent at the time of the review, we are happy to respond in follow-up comments as needed.

---

### Official Review · Reviewer_CG8r · 2022-10-26

**Confidence:** 3
**Correctness:** 3
**Technical Novelty And Significance:** 3
**Empirical Novelty And Significance:** 2
**Recommendation:** 5

**Clarity, Quality, Novelty And Reproducibility:**

Clarity: 8/10
Quality: 7/10
Novelty: 7/10
Reproducibility: 7/10 (if code released)

**Strength And Weaknesses:**

Strengths:
1. The idea of utilizing human-designed taxonomy prior in latent space learning is interesting.
2. The introduction of Hyperbolic space seems to lead to better structure than Euclidean space.


Weaknesses:
1. I feel that the experimental part is not enough to demonstrate the advantage of the proposed GPHLVM:
(a) There is only one task/dataset (whole-body support pose taxonomy) is used in the experiments. It is thus unclear whether the proposed method can be applied to other tasks as well.
(b) There are no other alternative approaches for comparison (only Gaussian counterpart for visual analysis).
(c) There is no quantitative comparison on some concrete tasks (e.g., pose classification) to demonstrate the practical benefit of the latent space.

2. An alternative design would be using a neural network to learn an embedding space. We can still apply the priors (i.e., Hyperbolic and taxonomy-aware priors) to the learned embedding space by designing various loss functions. However, this alternative approach will be more straightforward and efficient to implement. Unfortunately, we fail to find such discussion or comparison in the paper.

3. The proposed method hopes that the latent space complies with the human-designed taxonomy's hierarchical structure. However, as shown in the part of the experiments, when the human-designed taxonomy is coarse-grained, the performance of the proposed method is also limited.

4. The runtime information is missing from the text. It's unclear whether the proposed method is an efficient or computationally intractable one.

5. The current latent space is very low-dimensional. Could the authors provide more discussion on the benefits of learning a low-dimensional latent space? Also, it's interesting to know whether the proposed method can handle high-dimensional latent space as well.

6. How shall we interpret the pairwise distance matrices in Figure 2 and compare them with the graph distances in Figure 3. To be honest, it's not easy to tell which one is closer to Figure 3.

7. The interpolated motions in Figure 4 fail to impress me. The transition is not very natural and smooth. Moreover, it includes many physically-infeasible motions. In contrast, Figure 5 shows better performance. Since only motions generated by the proposed GPHLVM are shown, it is hard to demonstrate the advantage of the proposed GPHLVM, as other methods may generate similar or better performances.

**Summary Of The Paper:**

This paper proposes a method to learn an embedding space, which leverages the human-designed taxonomy, for robotics tasks. It extends Gaussian Process Latent Variable Models (GPLVM) to hyperbolic latent spaces and proposes GPHLVM. The motivation for using hyperbolic space is that distances grow exponentially when moving away from the origin, and the shortest paths between distant points tend to pass through it, resembling a continuous hierarchical structure. They propose graph-distance priors and back constraints to introduce taxonomy knowledge into the GPHLVM.

The experiments are conducted on whole-body support pose taxonomy. The authors analyze the learned Hyperbolic latent embedding space and compare it with the Euclidean one. They also show that the learned embedding enables motion interpolation in the latent space.

**Summary Of The Review:**

I feel that the experimental part is not enough to demonstrate the advantage of the proposed over existing methods. The practical usage is also unclear.

---

> ### Author Response · Authors · 2022-11-18
> **Response to comments of Reviewer CG8r - part 1**
>
> Thank you for taking the time to review our work! We are delighted to read that the idea of our paper is interesting. Below, we address some of the key concerns raised as part of the review. The corresponding changes are highlighted in blue in our paper.
>
> 1. We would like to draw the attention of the reviewers to the variety of datasets and taxonomy graphs used in our experiments and to the additional experiments that are reported in **Appendix G** of our paper, as mentioned in **Section 5**.
>
>     (a) **Variety of datasets and taxonomy graphs:** We would like to emphasize that our experiments considered three variations of the datasets of (Mandery et al., 2016) and (Langenstein, 2020) and three different taxonomy graphs: (i) the three first parts of **Section 5** consider 100 poses (72 standing and 28 kneeling poses) represented as 12-dimensional vectors composed of the positions of the human's feet and hands. This dataset is *unbalanced*. Each pose is identified with a node of the *original* whole-body support pose taxonomy (Borràs et al., 2017), i.e., the taxonomy graph is given by Fig. 1-right; (ii) the additional experiment presented in **Appendix G.2** involves 60 standing poses represented as 12-dimensional vectors containing the positions of the human's feet and hands. This dataset is *balanced* (5 pose per contact set). Each pose is identified with a node of the *standing part* of the whole-body support pose taxonomy, i.e., the taxonomy graph corresponds only to the *left part* of Fig. 1-right; (iii) The last part of **Section 5** and the accompanying **Appendix G.3** report results on a *different representation* of the 60 standing poses. Namely, to facilitate motion planning and test the ability of our model to deal with high-dimensional spaces, each pose is represented as a 44-dimensional vector of joint angles instead of a 12-dimensional vector of hands and feet positions. Last but not least, the last part of **Section 5** and **Appendix G.3** consider an *augmented* version of the whole-body support pose taxonomy, which explicitly distinguishes between left and right contacts. In other words, the number of nodes increases from 15 in the original taxonomy (see Fig. 1-right) to 32 in the augmented taxonomy., therefore both taxonomies are fundamentally different.
>
>     Therefore, we believe that our experiments successfully showcase the proposed GPHLVM for different datasets (standing and kneeling poses v.s. standing poses only, balanced and unbalanced cases, node features composed of hand and feet positions, and high-dimensional node features built from joint angles) and various taxonomy graphs (original whole-body support pose and standing pose taxonomies, as well as augmented whole-body standing pose taxonomy). Finally, due to space constraints (and to our already-lengthy supplementary material), we prefer to defer experiments with other robotic taxonomies to a future publication.
>
>     (b) **Alternative approaches:** Note that **Appendix G.4** compares the proposed GPHLVM with two additional baselines, namely a vanilla variational autoencoder (VAE) and a hyperbolic variant of VAEs with a hyperbolic latent space. We further discuss this comparison in our answer of the point 2 below.
>
>     (c) **Quantitative comparisons on concrete tasks:** Embeddings of taxonomy classes obtained via geometric representation learning are usually evaluated via two tasks, namely (i) hierarchical classification via label embeddings, and (ii) taxonomy extension (see, e.g., ([Marconi et al., 2020](https://proceedings.mlr.press/v108/marconi20a.html)). Accordingly, we evaluated the embeddings obtained via our GPHLVM via these two tasks in the subsection "Taxonomy expansion and unseen poses encoding" of **Section 5**. Namely, (i) the hierarchical classification is performed by encoding unseen poses and verifying that they are placed closed to the corresponding taxonomy class, and (ii) the taxonomy expansion is achieved by encoding poses corresponding to an unseen class and verifying that they are located at a sensible distance to the other taxonomy classes. Notice that the same two tasks were also evaluated in the additional experiments of **Appendix G**.
>
>     According to the reviewer's suggestion, we added a quantitative evaluation of these aforementioned tasks by reporting the average stress for the models with added previously-unseen class or poses in Table 1. Note that our GPHLVM displays lower stress values when placing unseen poses or taxonomy classes into the latent space.
>
>     (Marconi et al., 2020): Marconi, G. M., Rosasco, L., and Ciliberto, C. Hyperbolic manifold regression. In AISTATS (2020).

---

> > ### Author Response · Authors · 2022-11-18
> > **Response to comments of Reviewer CG8r - part 2**
> >
> > 2. We agree with the reviewer that the comparison with an alternative design based on neural network is relevant for our paper. As mentioned in **Section 5**, **Appendix G.4** includes such a comparison. We compared the proposed GPHLVM with two baselines: (i) a vanilla variational autoencoder (VAE) and (ii) a hyperbolic variant of the VAE whose latent space is hyperbolic, akin to [Mathieu et al., 2019](https://proceedings.neurips.cc/paper/2019/file/0ec04cb3912c4f08874dd03716f80df1-Paper.pdf). The taxonomy prior is included by adding the stress regularization of Eq. (10) when training the VAEs. Importantly, our GPHLVM consistently outperformed all VAE models. As shown in Fig. 12, the VAE models seem to only capture a global structure that separates standing from kneeling poses. Moreover, the average stress of the latent embeddings is higher for the VAE baselines (see Table 11). We refer the reviewers to **Appendix G.4** for further details.
> >
> >     Moreover, we believe that, due to their data efficiency, GPLVMs are well-adapted to the type of problems we tackle, where only few data are available. They also provide better encoding quality than deep learning-based methods, e.g., VAEs, especially in low-data regimes. Last but not least, GPLVMs also provide automatic uncertainty quantification, as opposed to VAEs.
> >
> >     We added a brief description of the results of **Appendix G.4** in **Section 5**.
> >
> >     (Mathieu et al., 2019): Mathieu, E., Lan, C. L., Maddison, C. J., Tomioka, R., and Teh, Y. W. Continuous Hierarchical Representations with Poincaré Variational Auto-Encoders. In NeurIPS (2019).
> >
> > 3. We agree with the reviewer that our method requires a well-designed taxonomy. Our work aims at leveraging the domain knowledge that researchers introduced in the design of robotics taxonomies. Therefore, ill-designed taxonomies will naturally lead to suboptimal embeddings. It is important to notice that this relates to common issues encountered with machine learning models such as handling mislabeled data during model training.
> >
> >     We would also like to emphasize that the proposed method goes beyond encoding the taxonomy: Namely, the fact that we embed features associated to a taxonomy in a *continuous* hyperbolic space allows us to, e.g., generate motions from an *originally-discrete* taxonomy.
> >
> > 4. As suggested by the reviewer, we added the runtime information in **Appendix G.5**.
> >
> > 5. In general, low-dimensional latent spaces are advantageous as (i) they are easily visualized (especially for dimension $Q=2$, and to some extent for $Q=3$), (ii) they are easy to work with, and (iii) operations in low-dimensional space are computationally cheaper than in higher-dimensional latent spaces, which may impact downstream applications such as motion generation. It is important to emphasize that, thanks to the geometry of the hyperbolic manifold, the proposed GPHLVM encodes complex taxonomy structures in low-dimensional latent spaces ($Q=2$ or $Q=3$) which inherits all the aforementioned advantages.
> >
> >     Yet, the proposed GPHLVM naturally handles higher-dimensional latent spaces, if required by the complexity of a particular taxonomy. This requires to employ the hyperbolic kernels for dimension $d>3$ as defined in [Grigoryan \& Noguchi, 1998](https://www.math.uni-bielefeld.de/~grigor/nog.pdf) for the squared-exponential case, and as introduced in [Jaquier et al., 2021](https://openreview.net/pdf?id=ovRdr3FOIIm) for the Matérn case. The training of the GPHLVM remains identical (see Section 3.2).

---

> > > ### Author Response · Authors · 2022-11-18
> > > **Response to comments of Reviewer CG8r - part 3**
> > >
> > > 6. The interpretation of the pairwise distance matrix is related to the stress values of Table 1, computed as the average of the stress loss of Eq. (10). As shown in Table 1, the hyperbolic embeddings obtained with stress regularization achieved lower stress values, i.e., their pairwise distance matrices are closer to the graph distance of Fig. 3. One can also observe that some parts of the distance matrices of the Euclidean embeddings differ strongly from Fig. 3. For instance, the bottom-left part of the distance matrix in the last row of Fig. 2c shows white zones indicating a distance $>=6$, which barely occurs in Fig. 3.
> > >
> > > 7. We agree with the reviewer on the limited quality of the motions displayed in Fig. 4. As discussed in **Section 5**, this is due to the lack of information about the type of contacts in the considered poses and to the fact that the poses are represented as the positions of the human's feet and hands, thus ignoring the feet and hands orientation. Despite this, we believe that Fig. 4 shows very interesting results, as the resulting motions are consistent with the taxonomy transitions. Moreover, we addressed the aforementioned limitations in the last part of **Section 5** ("Augmented taxonomy for enhanced trajectory generation") resulting in the improved performance displayed in **Figure 5**.
> > >
> > >     We followed the recommendation of this reviewer and added the trajectories generated in the latent space of the vanilla Euclidean GPLVM in **Appendix G.3** (Fig. 12), as well as the trajectories generated in the latent space of the VAE baselines in **Appendix G.4** (Fig. 15). Overall, the motions planned in the hyperbolic latent space of the proposed GPHLVM look more realistic than those planned in the Euclidean latent space of the vanilla GPLVM. As shown in Fig. 12, the motions planned in the Euclidean latent space sometimes result in unrealistic joint configurations and the same posture is associated with different types of contacts (see middle part of the motions). Moreover, as shown in Table 10, non-existing transitions arise more frequently when following trajectories generated by the Euclidean model. Similarly as the motions planned in the hyperbolic latent space of the proposed GPHLVM, the geodesics in the latent space the hyperbolic VAE result in realistic motions.
> > >
> > >     Comparisons between the transitions obtained with the GPHLVM and GPLVM are also available in **Appendices F.4, G.1, G.2, and G.3**. In contrast to GPHLVM, the Euclidean GPLVM often generates transitions that do not exist in the taxonomy. Interestingly, it also often uses more transitions than those originally needed, especially for embeddings of dimension $Q=2$.
> > >
> > > **Reproducibility:** The code related to the paper will be made publicly available shortly.

---

### Author Response · Authors · 2022-12-06
**Rebuttal discussion**

Dear Reviewers and AC,

We would be glad to have the Reviewers' feedback on our rebuttal answers and on the paper updates! We would also be happy to address and discuss further comments.

Best regards,
The Authors

---

### Decision · Program_Chairs · 2023-01-20

**Decision:**

Reject

**Justification For Why Not Higher Score:**

The experimental part  of the paper is not sufficient.  Furthermore, an at least a coarse comparison with other approaches to geodesics would be needed (NOT using hyperbolic latent spaces).  The use of hyperbolic spaces is not defended (but indeed used before).

**Justification For Why Not Lower Score:**

n/a

**Metareview: Summary, Strengths And Weaknesses:**

In this paper, robot taxonomies are embedded in GP-LVMs, extended to hyperbolic latent spaces, to enable geodesic distances.

The approach is interesting and the experimental results are OK.  But the paper does not compare with NN-based geodesics-based approaches, e.g. Tosi et al; Artavanidis et al, ICLR, 2018; Chen et al, ICML, 2020.  The comparison just with VAE and a hyperbolic VAE seems unjust, as better results exist, albeit not with hyperbolic latent spaces.   Advantages of hyperbolic latent spaces over other forms are not investigated.

The authors indicate very high training times for embedding in (relatively) low-dimensional latent spaces.  It is unclear how that generalises for more complex problems, where the inherent latent space is higher-dimensional yet lower-embedded.  Unfortunately, the rather restricted experiments give no clue to such issues.  Indeed, for the small experimental part, the results are not always convincing.